# A kernel test for quasi-independence

**Tamara Fernández**
Gatsby Unit
University College London
`t.a.fernandez@ucl.ac.uk`

**Wenkai Xu**
Gatsby Unit
University College London
`xwk4813@gmail.com`

**Marc Ditzhaus**
Department of Statistics
TU Dortmund University
`marc.ditzhaus@tu-dortmund.de`

**Arthur Gretton**
Gatsby Unit
University College London
`arthur.gretton@gmail.com`

## Abstract

We consider settings in which the data of interest correspond to pairs of ordered times, e.g, the birth times of the first and second child, the times at which a new user creates an account and makes the first purchase on a website, and the entry and survival times of patients in a clinical trial. In these settings, the two times are not independent (the second occurs after the first), yet it is still of interest to determine whether there exists significant dependence *beyond* their ordering in time. We refer to this notion as "quasi-(in)dependence". For instance, in a clinical trial, to avoid biased selection, we might wish to verify that recruitment times are quasi-independent of survival times, where dependencies might arise due to seasonal effects. In this paper, we propose a nonparametric statistical test of quasi-independence. Our test considers a potentially infinite space of alternatives, making it suitable for complex data where the nature of the possible quasi-dependence is not known in advance. Standard parametric approaches are recovered as special cases, such as the classical conditional Kendall's tau, and log-rank tests. The tests apply in the right-censored setting: an essential feature in clinical trials, where patients can withdraw from the study. We provide an asymptotic analysis of our test-statistic, and demonstrate in experiments that our test obtains better power than existing approaches, while being more computationally efficient.

## 1 Introduction

Many practical scientific problems require the study of events which occur consecutively in time. We focus here on the setting where event-times, $X$ and $Y$, are only observed if they are in the ordered relationship $X \leq Y$. This type of data is commonly known as *truncated data*, and, in particular, we say that $X$ is right-truncated by $Y$, or $Y$ is left-truncated by $X$. In clinical trails, for example, only patients still alive at the beginning of the study can be recruited, hence the recruitment times $X$ and the survival times $Y$ are ordered. In the field of insurance, a liability claim may be placed at a time $Y$ as a consequence of an incident at a time $X$. In e-commerce, the time $Y$ of first purchase by a new user may only happen after the time $X$ when the user registers with the website.

Our goal is to determine whether there exists an association between $X$ and $Y$ in the truncated data setting. Given that $X \leq Y$, the times $X$ and $Y$ clearly will not be independent (with the exception of trivial cases in which, for instance, $X$ and $Y$ have disjoint support). Thus, while it is not meaningful to test for statistical independence in the truncated setting, we can nevertheless still test for whether $X$ and $Y$ are uncoupled apart from the fact that $X \leq Y$, using the notion of *quasi-independence*. We will make this notion formal in Section 2.

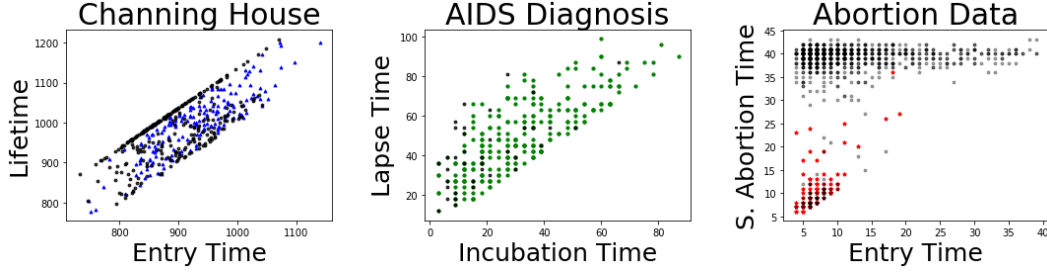

Figure 1: Channing House dataset: the x-axis shows the entry time to the retirement center; and the y-axis shows the right-censored lifetimes . Events are censored by withdrawal from the center or study finishes at July 1, 1975. AIDS dataset: the x-axis shows the incubation time X; and the y-axis shows the censored lapse time Y, measured from infection to recruitment time. Events are censored by death or left the study. Infected patients were recruited in the study only if they developed AIDS within the study period, therefore, in this dataset, the incubation time X does not exceed the lapse time Y. Abortion dataset: the x-axis shows the time to enter the study; and the y-axis shows the right-censored time for spontaneous abortion. Events are censored due to life birth and induced abortions. All censored times are marked in dark.

Testing for an association between ordered $X$ and $Y$ may be important in making business/medical decisions. In the setting of clinical trials, it is important to ensure that survival times are as "independent" as possible from recruitment times, in order to avoid bias in the recruitment process. In e-commerce, it may be of interest to test whether the purchase time for an item, such as a swimsuit, depends on the registration time, to determine seasonal effects on consumer behaviour and refine advertising strategies. In statistical modelling, a common working assumption is that $X$ and $Y$ are independent, but can only be observed when $X \leq Y$ holds: see e.g, [18, 33, 38], and [22, Chapter 9]. The independence assumption can be weakened to quasi-independence, which is testable, and under which typical methods are still valid [22, 24, 35, 36, 37, 38].

Our tests apply in the setting where $Y$ is right-censored. This is a very common scenario in real-world applications, particularly in clinical trials, where patients may withdraw from the study before their event of interest is observed. In the e-commerce example, there may be registered users that have not yet made a purchase when the study ends. Formally, the data corresponds to the triple $(X, T, \Delta)$, where $T = \min\{C, Y\}$ is the minimum between the survival time $Y$ of a given patient, and the time $C$ at which said patient leaves the study (or the study ends), and $\Delta = \mathbb{1}_{\{T=Y\}}$. Given the truncated data setting, we have further that $X \leq \min\{Y, C\}$. We emphasise that quasi-independence and right-censoring are very different data properties. Quasi-independence is a deterministic hard constraint $(X \leq Y)$, while right-censoring is a stochastic property of the data (incomplete observations).

Quasi-independence has been widely studied in the statistics community, including for right-censored data: we provide a brief review below (more detailed descriptions of relevant concepts and methods will be provided in subsequent sections).

In this work, we propose a non-parametric statistical test for quasi-independence, which applies under right censoring. Our test statistic is a nonparametric generalisation of the log-rank test proposed by [7], where the departure from the null is characterised by functions in a reproducing kernel Hilbert space (RKHS). Consequently, we are able to straightforwardly detect a very rich family of alternatives, including non-monotone alternatives. Our test generalises statistical tests of independence based on the Hilbert-Schmidt Independence Criterion [16]; which were adapted to the right-censoring setting in [8, 28]. Due to the additional correlations present in the test statistic under quasi-independence, however, we will require new approaches in our analysis of the consistency and asymptotic behaviour of our test statistic, compared with these earlier works.

In Section 2, we introduce the notion of quasi-independence. We next propose an RKHS statistic to detect this quasi-independence, and its finite sample estimate from data. We contrast the statistic for quasi-independence with the analogous RKHS statistic for independence, noting the additional sample dependencies on account of the left-truncation. Next, in Section 3, we generalise the quasi-independence statistics to account for the presence of right-censored observations. In Section 4, we provide our main theoretical results: an asymptotic analysis for our test statistic, and a guarantee of consistency under the alternative. In order to determine the test threshold in practice, we introduce

a Wild Bootstrap procedure to approximate the test threshold. In Section 5 we give a detailed empirical evaluation of our method. We begin with challenging synthetic datasets exhibiting periodic quasi-dependence, as would be expected for example from seasonal or daily variations, where our approach strongly outperforms the alternatives. Additionally, we show our test is consistently the best test in data-scenarios in which the censoring percentage is relatively high, see Figure 6. Next, we apply our test statistic to three real-data scenarios, shown in Figure 1: a survival analysis study for residents in the Channing House retirement community in Palo Alto, California [18]; a study of transfusion-related AIDS [24]; and a spontaneous abortion study [26]. For this last dataset, our general-purpose test is able to detect a mode of quasi-dependence discovered by a model that exploits domain-specific knowledge, but not found by alternative general-purpose testing approaches. This was a particular challenge due to the large percentage of censored observations in the abortion dataset; see Figure 6. Proofs of all results are given in the Appendix.

## 2  Quasi-independence

Our goal is to infer the null hypothesis of quasi-independence between $X$ and $Y$. Formally, this null hypothesis is characterised as

$$H_0 : \pi(x,y) = \widetilde{F}_X(x)\widetilde{S}_Y(y), \quad \text{for all } x \leq y, \tag{1}$$

where $\pi(x,y) = \mathbb{P}(X \leq x, Y \geq y)$, and $\tilde{F}_X(x)$ and $\tilde{S}_Y(y)$ are functions that only depend on $x$ and $y$, respectively. In case of independent $X$ and $Y$, $\widetilde{F}_X(x)$ and $\widetilde{S}_Y(y)$ coincide with $F_X(x) = \mathbb{P}(X \leq x)$ and $S_Y(y) = \mathbb{P}(Y \geq y)$, but in general they may differ. For simplicity, $X$ and $Y$ are assumed continuously distributed on $\mathbb{R}_+$, and $f_{XY}$, $f_X$ and $f_Y$ denote the joint density and the corresponding marginals, and $f_{Y|X=x}$ denotes the conditional density of $Y$ given $X = x$.

To simplify the notation, we suppose throughout that $X \leq Y$ always holds, and thus write $\pi(x,y) = \mathbb{P}(X \leq x, Y \geq y)$ instead of $\pi(x,y) = \mathbb{P}(X \leq x, Y \geq y | X \leq Y)$, as $\mathbb{P}(X \leq Y) = 1$. We remark, however, that the ordering $X \leq Y$ can be ensured by considering a conditional probability space given $X \leq Y$, and restricting calculations of probabilities, expectation etc. to this space; see [2, 7, 34].

The notion of quasi-independence must not be confused with the notion of independent increments, i.e., $X \perp (Y - X)$. For instance, generate $X$ and $Y$ such that $X \leq Y$ by sampling i.i.d. uniform random variables, say $(U_1, U_2)$, in the interval $(0, 1)$, and make $X = U_1$ and $Y = U_2$ for the first pair $(U_1, U_2)$ such that $U_1 \leq U_2$. It can be verified that this construction leads to quasi-independent random variables $(X, Y)$, but $X$ and $Y - X$ are not independent as the distribution of $Y - X$ is constrained by how large the original value of $X$ was. The larger $X$ is, the smaller is the value of $Y - X$.

In [7], the authors propose to measure quasi-independence by using a log-rank-type test-statistic which estimates $\int_{x \leq y} \omega(x,y)\rho(x,y)dxdy$, where

$$\rho(x,y) = -\pi(x,y)\frac{\partial^2\pi(x,y)}{\partial x \partial y} + \frac{\partial\pi(x,y)}{\partial x}\frac{\partial\pi(x,y)}{\partial y}, \; x \leq y. \tag{2}$$

The function $\rho$ is originally inspired by the odds ratio proposed by [1] (notwithstanding that $\rho$ is here a difference, rather than a ratio). Under the assumption of quasi-independence, $\rho = 0$, and thus $\int_{x \leq y} \omega(x,y)\rho(x,y)dxdy = 0$. Nevertheless, it may be that $\int_{x \leq y} \omega(x,y)\rho(x,y)dxdy = 0$ even if the quasi-independence assumption is not satisfied, since the quantity depends on the function $\omega$: for instance, it is trivially zero when $\omega = 0$. To avoid choosing a specific weight function $\omega$, we optimise over a class of weight functions, taking an RKHS approach,

$$\Psi = \sup_{\omega \in B_1(\mathcal{H})} \int_{x \leq y} \omega(x,y)\rho(x,y)dxdy, \tag{3}$$

where $B_1(\mathcal{H})$ is the unit ball of a reproducing kernel Hilbert space $\mathcal{H}$ with bounded measurable kernel given by $\mathfrak{K} : \mathbb{R}_+^2 \times \mathbb{R}_+^2 \to \mathbb{R}$. We refer to the measure $\Psi^2$ as *Kernel Quasi-Independent Criterion (KQIC)*. It can easily be verified that $\Psi \geq 0$; and, if $X$ and $Y$ are quasi-independent, then $\Psi = 0$. For $c_0$-universal kernels [30], we have that $\Psi = 0$ if and only if $X$ and $Y$ are quasi-independent: see Theorem 4.2.

Given the i.i.d. sample $((X_i, Y_i))_{i \in [n]}$, we can estimate $\Psi$ via $\Psi_n$, defined as

$$\Psi_n = \sup_{\omega \in B_1(\mathcal{H})} \left( \frac{1}{n} \sum_{i=1}^{n} \omega(X_i, Y_i) \widehat{\pi}(X_i, Y_i) - \frac{1}{n^2} \sum_{i=1}^{n} \sum_{k=1}^{n} \omega(X_i, Y_k) \mathbb{1}_{\{X_k \leq X_i < Y_k \leq Y_i\}} \right) \quad (4)$$

where $\widehat{\pi}(x, y) = \frac{1}{n} \sum_{m=1}^{n} \mathbb{1}_{\{X_m \leq x, Y_m \geq y\}}$, and notice that $\widehat{\pi}(x, y)$ estimates $\pi(x, y)$. Using a reproducing kernel $\mathfrak{K}$ that factorises, we obtain a simple expression for $\Psi_n^2$:

**Proposition 2.1.** *Consider* $\mathfrak{K}((x, y), (x', y')) = K(x, x')L(y, y')$. *Then*

$$\Psi_n^2 = \frac{1}{n^2} trace(\boldsymbol{K} \widehat{\boldsymbol{\pi}} \boldsymbol{L} \widehat{\boldsymbol{\pi}} - 2 \boldsymbol{K} \widehat{\boldsymbol{\pi}} \boldsymbol{L} \boldsymbol{A}^\intercal + \boldsymbol{K} \boldsymbol{A} \boldsymbol{L} \boldsymbol{A}^\intercal)$$

*where* $\boldsymbol{K}$, $\boldsymbol{L}$, *and* $\boldsymbol{A}$ *are* $n \times n$*-matrices with entries given by* $\boldsymbol{K}_{ik} = K(X_i, X_k)$, $\boldsymbol{L}_{ik} = L(Y_i, Y_k)$ *and* $\boldsymbol{A}_{ik} = \mathbb{1}_{\{X_k \leq X_i < Y_k \leq Y_i\}}/n$, *and* $\widehat{\boldsymbol{\pi}}$ *is a diagonal matrix with entries* $\widehat{\boldsymbol{\pi}}_{ii} = \widehat{\pi}(X_i, Y_i)$.

We remark that the previous expression is similar in form to the Hilbert Schmidt Independence Criterion [14]. In particular, for empirical distributions, $\mathrm{HSIC}(\widehat{F}_{XY}, \widehat{F}_X \widehat{F}_Y) = \frac{1}{n^2} trace(\boldsymbol{K} \boldsymbol{H}^\intercal \boldsymbol{L} \boldsymbol{H})$ with $\boldsymbol{H} = \boldsymbol{I}_n - \frac{1}{n} \boldsymbol{1}_n \boldsymbol{1}_n^\intercal$, whereas our test-statistic can be rewritten as $\Psi_n^2 = \frac{1}{n^2} trace(\boldsymbol{K} \tilde{\boldsymbol{H}}^\intercal \boldsymbol{L} \tilde{\boldsymbol{H}})$ with $\tilde{\boldsymbol{H}} = (\widehat{\boldsymbol{\pi}} - \boldsymbol{A}^\intercal)$. Note that $\tilde{\boldsymbol{H}}$ is much more complex than $\boldsymbol{H}$, being a random matrix where each entry depends on all the data points. As we will see, this issue makes the asymptotic analysis in our case much more challenging; by contrast, the asymptotic distribution for HSIC can be readily obtained using standard results on U-statistics [16, 4].

Our test can be understood as a generalisation of the log-rank test proposed by [7], where instead of considering a single log-rank test with a specific weight function, we consider the supremum over a collection of log-rank tests with weights in $B_1(\mathcal{H})$. By choosing a sufficiently rich RKHS, for example the RKHS induced by the exponentiated quadratic kernels, we are able to ensure power against a broad family of alternatives. Conversely, simple kernels can recover classical parametric tests such as the aforementioned log-rank tests. As explained by [7, Equation 7], the simplest possible (constant) function space recovers the well-known conditional Kendall's tau:

**Proposition 2.2** (Recovering conditional Kendall's tau). *Consider* $\mathfrak{K} = 1$, *then* $\Psi_n^2 = K_a^2/n^2$, *where* $K_a = \sum_{i<k} \mathbb{1}_{\{X_i \vee X_k \leq Y_i \wedge Y_k\}} \mathrm{sign}\left((X_i - X_k)(Y_i - Y_k)\right)$ *is an empirical estimator of the conditional Kendall's tau.*

## 3 Right-censoring

In clinical trails, for example, patients might withdraw from the study before observing the time $Y$ of interest leading to so-called right-censored data. To model this kind of data, we introduce additionally the random censoring time $C$. The data correspond now to i.i.d. samples $((X_i, T_i, \Delta_i))_{i \in [n]}$, where $T_i = \min\{Y_i, C_i\}$ is the observation time, and $\Delta_i = \mathbb{1}_{\{T_i = Y_i\}}$ is the corresponding censoring status. In particular, if $\Delta_i = 0$, we only observe the censoring time $T_i = C_i$, and not the time of interest $Y_i$. Throughout, we assume that $X_i < T_i$ always holds, to reflect the natural ordering of the times, i.e. first recruitment and second the event of interest or the withdrawal from the study. As for the uncensored setting, $X$, $Y$ and $C$ are supposed to be continuously distributed on $\mathbb{R}_+$. Our results are valid under the standard non-informative censoring assumption:

**Assumption 3.1.** *The censoring times are independent of the survival times given the entry times, i.e.,* $C_i \perp Y_i | X_i$.

Standard notation for marginal, joint and conditional densities will be used: for instance, $f_C$, $f_{XT}$ and $f_{Y|X=x}$, are the marginal density of $C$, the joint density of $X$ and $T$, and the conditional density of $Y$ given $X = x$, respectively. Moreover, $S_Y$ denotes the survival function of $Y$, defined as $S_Y(y) = \mathbb{P}(Y \geq y)$ and $S_{C|X=x}(y) = \mathbb{P}(Y \geq y | X = x)$ is the conditional survival function of $Y$ given $X = x$. Under Assumption 3.1 we have $S_{T|X=x}(y) = S_{Y|X=x}(y) S_{C|X=x}(y)$.

The null hypothesis of *quasi-independence* is formulated, for the right-censored setting, as

$$H_0 : f_{XY}(x, y) = \tilde{f}_X(x) \tilde{f}_Y(y), \quad \text{for all } x \leq y, \text{ s.t. } S_{T|X=x}(y) > 0. \quad (5)$$

As with the uncensored case, $\tilde{f}_X$ and $\tilde{f}_Y$ are not necessarily equal to the marginal densities $f_X$ and $f_Y$. The additional condition $S_{T|X=x}(y) > 0$ ensures that the pair $(x, y)$ is actually observable despite the censoring. The statistic $\Psi$ from Equation (3) is then extended to the censored setting,

$$\Psi_c = \sup_{\omega \in B_1(\mathcal{H})} \int_{x \leq y} \omega(x, y) \rho^c(x, y) dx dy \geq 0,$$

$$\text{where } \rho^c(x, y) = -\pi^c(x, y) \frac{\partial^2}{\partial x \partial y} \pi_1^c(x, y) + \frac{\partial \pi^c(x, y)}{\partial x} \frac{\partial \pi_1^c(x, y)}{\partial y},$$

and $\pi_1^c(x, y) = \mathbb{P}(X \leq x, T \geq y, \Delta = 1)$ and $\pi^c(x, y) = \mathbb{P}(X \leq x, T \geq y)$ for $x \leq y$.

**Proposition 3.2.** *We have $\Psi_c = 0$ if the null hypothesis $H_0$ of quasi-independence is fulfilled.*

The (updated) estimator for the (new) Kernel Quasi Independent Criterion $\Psi_c$ is defined by

$$\Psi_{c,n} = \sup_{\omega \in B_1(\mathcal{H})} \left( \frac{1}{n} \sum_{i=1}^n \Delta_i \omega(X_i, T_i) \widehat{\pi}^c(X_i, T_i) - \frac{1}{n^2} \sum_{i=1}^n \sum_{k=1}^n \Delta_k \omega(X_i, T_k) \mathbb{1}_{\{X_k \leq X_i < T_k \leq T_i\}} \right), \tag{6}$$

where $\widehat{\pi}^c(x, y) = \frac{1}{n} \sum_{m=1}^n \mathbb{1}_{\{X_m \leq x, T_m \geq y\}}$ is the natural estimator for $\pi^c(x, y)$. In the uncensored case, i.e. $\Delta = 1$ with probability 1, the new KQIC $\Psi_c$ and its estimator $\Psi_{c,n}$ collapse to the respective quantities $\Psi$ and $\Psi_n$ from Section 2. Moreover, the estimator $\Psi_{c,n}$ can be simplified for factorising kernels:

**Proposition 3.3.** *Consider $\mathfrak{K}((x, y), (x', y')) = K(x, x') L(y, y')$, then*

$$\Psi_{c,n}^2 = \frac{1}{n^2} trace(\boldsymbol{K} \widehat{\boldsymbol{\pi}}^c \tilde{\boldsymbol{L}} \widehat{\boldsymbol{\pi}}^c - 2\boldsymbol{K} \widehat{\boldsymbol{\pi}}^c \tilde{\boldsymbol{L}} \boldsymbol{B}^\intercal + \boldsymbol{K} \boldsymbol{B} \boldsymbol{L} \boldsymbol{B}^\intercal) \tag{7}$$

*where $\boldsymbol{K}_{ik} = K(X_i, X_k)$, $\tilde{\boldsymbol{L}}_{ik} = \Delta_i \Delta_k L(T_i, T_k)$, $\boldsymbol{B}_{ik} = \mathbb{1}_{\{X_k \leq X_i < T_k \leq T_i\}}/n$, and $\boldsymbol{\pi}^c$ is a diagonal matrix where $\widehat{\boldsymbol{\pi}}_{ii}^c = \widehat{\pi}(X_i, T_i)$.*

# 4 Asymptotic analysis and wild bootstrap

We now present our main two theoretical results. First, we establish the asymptotic null distribution of our statistic $n\Psi_{c,n}^2$.

**Theorem 4.1.** *Assume $\mathfrak{K}$ is bounded. Then, under the null hypothesis, $n\Psi_{c,n}^2 \xrightarrow{\mathcal{D}} \mu + \mathcal{Y}$, where $\mu$ is a positive constant, $\mathcal{Y} = \sum_{i=1}^\infty \lambda_i(\xi_i^2 - 1)$, $\xi_1, \xi_2, \ldots$ are independent standard normal random variables, and $\lambda_1, \lambda_2, \ldots$ are non-negative constants depending on the distribution of the random variables $(X, Y, C)$ and the kernel $\mathfrak{K}$.*

To verify Theorem 4.1, we show that the scaled version of our statistic, $n\Psi_{c,n}^2$, can be expressed under the null hypothesis as the sum of a certain V-statistic and an asymptotically vanishing term. To find this representation, we write our test-statistic as a double integral with respect to a martingale, and use martingale techniques, and the results introduced in [10], to show that the error incurred by replacing certain quantities by their population versions vanishes as the number of data points grows to infinity. The full proof is provided in Appendix C. We next establish conditions for consistency of the test under the alternative.

**Theorem 4.2.** *Let $\mathfrak{K}$ be a bounded, $c_0$-universal kernel [30]. Then $\Psi_{c,n}^2 \to \Psi_c^2$ in probability. Moreover, whenever the null hypothesis is violated, $\Psi_c^2$ is positive, implying that $n\Psi_{c,n}^2 \to \infty$ in probability,.*

We remark that the factorised kernel $\mathfrak{K}((x, y), (x', y')) = K(x, x') L(y, y')$ must be $c_0$-universal in the product space, which is true for instance when $K$ and $L$ are exponentiated quadratic kernels [11]. In the case of independence testing, a simpler condition on the kernel can be used, where kernels are required to be individually characteristic to their respective domains [13]. Whether this simple condition can be generalised to the quasi-independence setting remains a topic for future work.

The consistency result in Theorem 4.2 relies on the interpretation of the test statistic $\Psi_{c,n}$ and the KQIC $\Psi_c$, as the Hilbert space distances of the embeddings of certain positive measures. These

distances measure the degree of (quasi)-dependence. Under the $c_0$-universality assumption, the embedding of finite signed measures are injective [30], which, in our case, implies $\rho^c(x,y) = 0$ for almost all $x \le y$. It remains to prove that quasi-independence holds. To show this, we first note that $\rho^c(x,y) = 0$ implies

$$\frac{\partial^2 \pi_1^c(x,y)}{\partial x \partial y} = \frac{1}{\pi^c(x,y)} \frac{\partial \pi^c(x,y)}{\partial x} \frac{\partial \pi_1^c(x,y)}{\partial y}, \tag{8}$$

and that $\frac{\partial^2 \pi_1^c(x,y)}{\partial x \partial y} = S_{C|X=x}(y) f_{XY}(x,y)$. By carefully analysing Equation (8) we find an explicit decomposition of $f_{XY}(x,y)$ into the product of two functions only depending on $x$ and $y$, respectively, from which quasi-independence follows. A detailed proof is provided in Appendix D.

As noted above, the eigenvalues $\lambda_i$ in Theorem 4.1 — and thus, the limit distribution of our test statistic under the null hypothesis — depend on the unknown distribution of $(X, Y, C)$. For this reason, we propose to approximate the limit null distribution and its $(1 - \alpha)$-quantile $q_\alpha$ of $\mu + \mathcal{Y}$ using a wild bootstrap approach. This strategy is well-established for $V$- and $U$-statistics [6], and has successfully been applied in scenarios, similar to the present one, where the test statistic behaves asymptotically as a $V$-statistic [8, 9].

To introduce the wild bootstrap counterpart $\Psi_{c,n}^{\mathrm{WB}}$ of our statistic $\Psi_{c,n}$, let $W_1, \ldots, W_n$ be independent and identically distributed Rademacher random variables, and define the $n \times n$ matrix $\boldsymbol{K}^W$ with entries $\boldsymbol{K}_{ik}^W = W_i W_k K(X_i, X_k)$. Then,

$$(\Psi_{c,n}^{\mathrm{WB}})^2 = \frac{1}{n^2} \mathrm{trace}(\boldsymbol{K}^W \widehat{\boldsymbol{\pi}}^c \tilde{\boldsymbol{L}} \widehat{\boldsymbol{\pi}}^c - 2\boldsymbol{K}^W \widehat{\boldsymbol{\pi}}^c \tilde{\boldsymbol{L}} \boldsymbol{B}^{\intercal} + \boldsymbol{K}^W \boldsymbol{B} \tilde{\boldsymbol{L}} \boldsymbol{B}^{\intercal}).$$

We propose the test $\varphi_n^{\mathrm{WB}} = \mathbb{1}\{\Psi_{c,n}^2 > q_\alpha^{\mathrm{WB}}\}$ to infer $H_0$, where $q_\alpha^{\mathrm{WB}}$ denotes the $(1 - \alpha)$-quantile of $\Psi_{c,n}^{\mathrm{WB}})^2$ given the observations $((X_i, \Delta_i, T_i))_{i \in [n]}$.

## 5 Experiments

We perform synthetic experiments followed by real data applications. In the first set of synthetic examples, we replicate the settings studied in [3], where Gaussian copula models were used to create dependencies between $X$ and $Y$. In the second synthetic experiment, we investigate distribution functions $f_{Y|X=x}$ that have a periodic dependence on $x$. We then apply our tests to real-data scenarios such as those studied in [7] and [26].

**Methods** We implement the proposed quasi-independence test based on the test-statistic **KQIC** given in Equation (7). The kernels are chosen to be Gaussian with bandwidth optimised by using approximate test power [15, 21]. See Appendix G for details. Competing approaches include: **WLR**, the weighted log-rank test proposed in [7], with weight function chosen equal to $n\widehat{\pi}^c(x,y)$;[1] **WLR_SC**, the weighted log-rank test proposed in [7], with weight function chosen as suggested by the authors, i.e., $W(x,y) = \int_0^x \widehat{S}_{C_R}((y-u)-)^{-1}\widehat{\pi}^c(du,y)$, where $\widehat{S}_{C_R}$ is the Kaplan-Meier estimator associated to the data $((C_i - X_i, 1 - \Delta_i))_{i=1}^n$; **M&B**, the conditional Kendall's tau statistic modified to incorporate censoring as proposed in [25]; and **MinP1** and **MinP2**, the "minimal p-value selection" tests proposed in [3], which rely on permutations of the observed pairs. A review of these approaches can be found in Appendix F. For the synthetic experiments, we recorded the rejection rate over 200 trials. The wild-bootstrap size for **KQIC** and the permutation size for **MinP1**, **MinP2** are set to be 500.

**Monotonic Dependency** The first synthetic example from [3] is generated as follows: $X \sim \mathrm{Exp}(5)$ and $Y \sim \mathrm{Weibull}(3, 8.5)$; $(X, Y)$ are then coupled via a 2-dimensional Gaussian copula model with correlation parameter $\rho$. The censoring variable is set to be exponentially distributed and truncation applies. With the copula construction, the magnitude of the correlation parameter $\rho$ is a fair indicator of the degree of dependence, with $\rho = 0$ denoting independence. Rejection rates are reported in Table 1. At $\rho = 0$, the null hypothesis holds, and the rejection rates refer to the Type-I error. All the tests achieve a correct Type-I error around a test level $\alpha = 0.05$. For $\rho \ne 0$, the alternative holds, and the rejection rates correspond to test power (the higher the better). The highest value is in bold. Test results w.r.t. different censoring rates can be found in the Appendix. Overall, our method outperforms all competing approaches.

| $\rho$ | -0.4 | -0.2 | 0.0 | 0.2 | 0.4 | -0.4 | -0.2 | 0.0 | 0.2 | 0.4 |
|---|---|---|---|---|---|---|---|---|---|---|
| KQIC | **0.93** | **0.46** | 0.06 | **0.42** | **0.86** | **0.99** | **0.67** | 0.05 | **0.63** | **1.00** |
| WLR | 0.80 | 0.33 | 0.10 | 0.18 | 0.66 | 0.94 | 0.52 | 0.06 | 0.32 | 0.94 |
| WLR_SC | 0.85 | 0.42 | 0.03 | 0.24 | 0.74 | 0.93 | 0.53 | 0.06 | 0.43 | 0.99 |
| M&B | 0.64 | 0.22 | 0.02 | 0.16 | 0.74 | 0.94 | 0.28 | 0.03 | 0.42 | 0.92 |
| MinP1 | 0.58 | 0.12 | 0.03 | 0.17 | 0.62 | 0.84 | 0.12 | 0.10 | 0.34 | 0.84 |
| MinP2 | 0.33 | 0.04 | 0.06 | 0.10 | 0.28 | 0.56 | 0.08 | 0.08 | 0.28 | 0.52 |

Table 1: Rejection rates for monotonic dependency models based on Gaussian copula, with $n = 100$ on the left; $n = 200$ on the right; $\alpha = 0.05$; censoring rate: $50\%$.

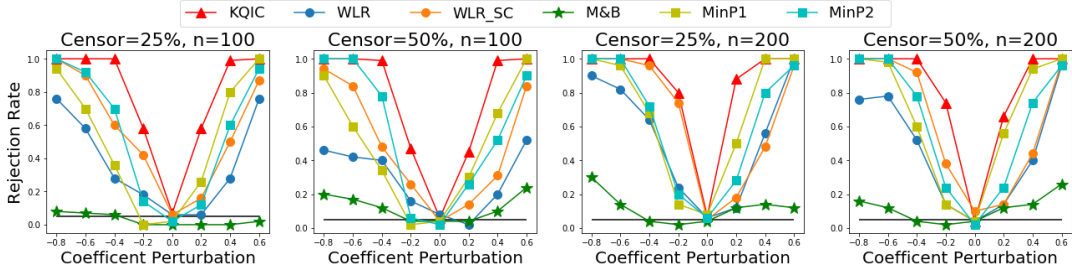

Figure 2: Rejection rate for V-shape Gaussian copula model

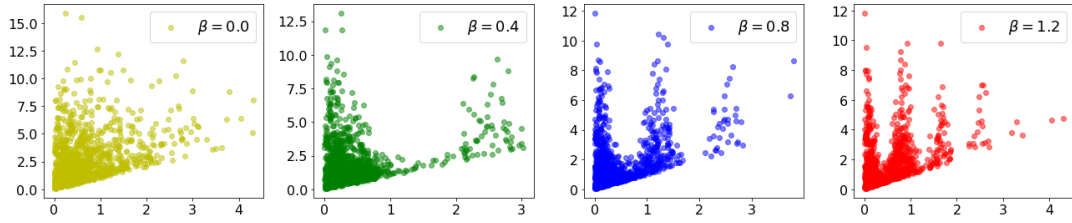

Figure 3: Samples from Periodic Dependency Model w.r.t. Frequency Coefficient $\beta$

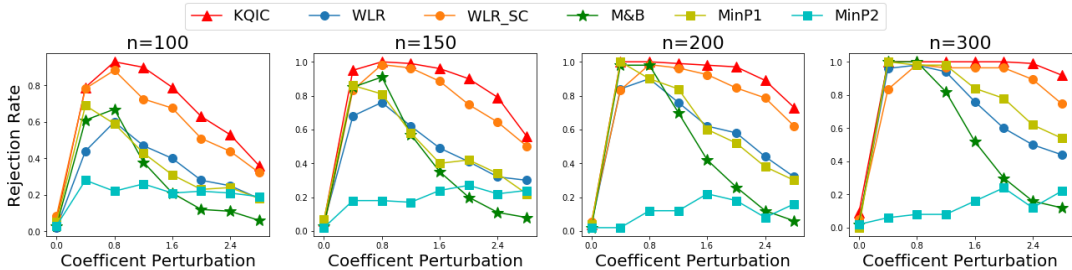

Figure 4: Rejection Rate for Periodic Dependency Model with $25\%$ data censored.

**V-shaped Dependency** A synthetic example [3], in which the authors compare the behaviour of their tests against the conditional Kendall's tau test of [25] in detecting non-monotonic dependencies. The following V-shaped dependency structure applies: $X \sim \text{Weibull}(0.5, 4)$; $Y \sim \text{Uniform}[0, 1]$; $(X, |Y - 0.5|)$ is coupled via the 2-dimensional Gaussian copula with correlation coefficient $\rho$ as above. Exponential censoring and truncation apply. Rejection rates are plotted against the perturbation of correlation coefficient $\rho$ in Figure 2, where KQIC outperforms competing methods.

**Periodic Dependency** Apart from the V-shaped dependencies studied in [3], we investigate more complicated non-monotonic dependencies structures. The data are generated with a periodic dependency structure, $X \sim \text{Exp}(1)$; $Y|X \sim \text{Exp}(e^{\cos(2\pi\beta X)})$. The coefficient $\beta$ controls the frequency of

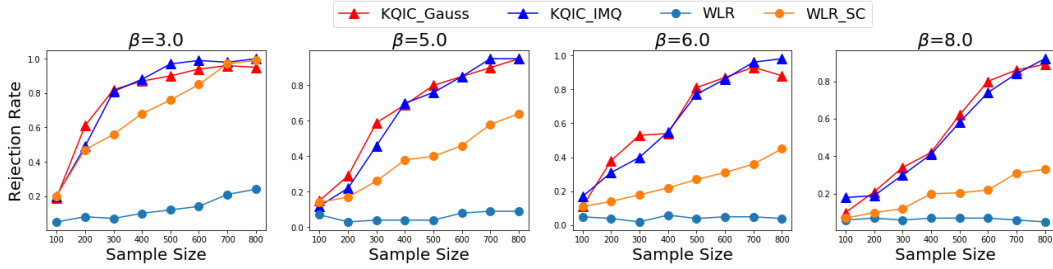

Figure 5: Rejection rate for high frequency dependency, with $\alpha = 0.05$, 40% data censored

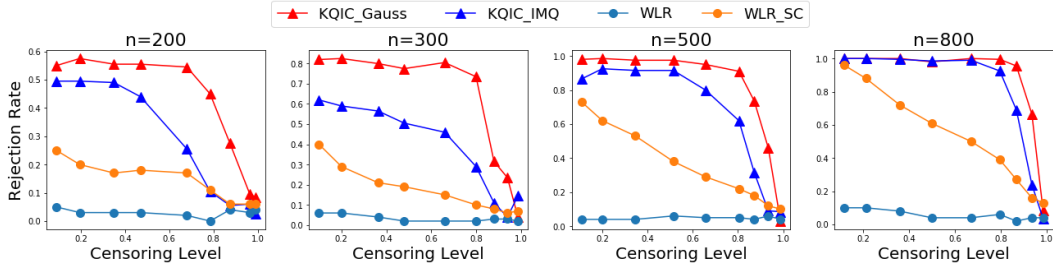

Figure 6: Rejection rate for periodic dependencies ($\beta = 5.0$), with $\alpha = 0.05$ and 200 trials.

the dependence. A set of examples with different parameters $\beta$ is shown in Figure 3, with $\beta = 0$ implying independence. Further details are discussed in Appendix G.3.

Examining the results in Figure 4, we see that our method outperforms competing approaches. Unlike the correlation coefficient $\rho$ in Gaussian copula models, the coefficient $\beta$ does not directly imply the "amount" of dependence; rather, a higher $\beta$ indicates a more "difficult" problem. Thus, as anticipated, power drops for large values of $\beta$, and the effect is more apparent at low sample sizes. Note in particular that the permutation based tests [3] are more affected by an increase in frequency at which dependence occurs, while our test shows a more robust behaviour.

**High Frequency Dependency** In the period dependency problem above , the parameter $\beta$ controls the frequency of sinusoidal dependence. At a given sample size, the dependence becomes harder to detect as the frequency $\beta$ increases. We visually show this in Appendix G.3. For problems with high frequency dependence, a larger sample size is required.

When the sample size increases, KQIC is able to successfully reject the null at relatively high frequencies (large $\beta$), as shown in Figure 5. At lower frequencies $\beta = 3.0$, WLR_SC has similar test power as KQIC. As the problem gets harder with larger $\beta$, KQIC outperforms WLR_SC. The IMQ kernel has similar test power as the Gaussian kernel on this example. We report the Type-I error that is well controlled in Appendix G.3 Table 5.

**Censoring level** We investigate how our test is affected by the censoring level, in particular when the censoring percentage increases. We analyse performance under both the null and alternative hypotheses. The Type-I error is well controlled for KQIC and details are reported in Appendix G.5.

Under the alternative hypothesis, in Figure 6, we show the rejection rate w.r.t. different censoring percentages and fixed sample size. This is done in our periodic dependency setting. From the plot, we see that KQIC with Gaussian and IMQ kernels is more robust to censoring, with test power starting to drop at 85% of censoring for sample size = 800. WLR_SC is strongly affected by censoring. WLR is not capable of detecting $H_1$ in this hard problem with high frequency.

In addition, we study the test behaviour with dependent censoring, since in Assumption 3.1, only conditional independence $Y \perp C|X$ is required [7]. Detailed results are reported in Appendix G.4.

**Computational cost** Our proposed test, implemented as described in Appendix E, has a significantly lower runtime when compared with the competing permutation approaches. M&B implements the conditional Kendall's tau statistic, which has a closed-form expression for the null distribution, therefore its runtime is lowest of all. See Appendix G for details.

| (p-value) | Channing House | | | AIDS | Abortion Times | | |
|---|---|---|---|---|---|---|---|
| | Combined | Male | Female | | Combined | Control | Treatment |
| KQIC_Gauss | 0.072 | 0.012 | 0.566 | 0.030 | 0.014 | 0.440 | 0.028 |
| KQIC_IMQ | 0.078 | 0.022 | 0.414 | 0.010 | 0.032 | 0.158 | 0.048 |
| WLR | 0.058 | 0.016 | 0.444 | 0.035 | 0.408 | 0.868 | 0.748 |
| WLR_SC | 0.086 | 0.020 | 0.422 | 0.030 | 0.511 | 0.674 | 0.450 |
| MinP1 | 0.084 | 0.036 | 0.396 | 0.012 | 0.584 | 0.584 | 0.452 |
| MinP2 | 0.198 | 0.426 | 0.118 | 0.406 | 0.694 | 0.572 | 0.346 |
| M&B | 0.178 | 0.199 | 0.495 | 0.010 | 0.712 | 0.693 | 0.752 |
| % Events | 0.379 | 0.474 | 0.354 | 0.875 | 0.094 | 0.069 | 0.098 |

Table 2: Real data, with marked results contradicting and supporting the scientific literature.

**Real Data Experiment** We consider three real data scenarios: **Channing House** [18]: contains the recorded entry times and lifetimes of 461 patients (97 men and 364 women). Among them, 268 subjects withdrew from the retirement center, yielding to a censoring proportion of 0.62. The data are naturally left truncated, as only patients who entered the center are observed; **AIDS** [24]: the data contain the incubation time and lapse time, measured from infection to recruitment time, for 295 subjects. A censoring of proportion of 0.125 occurs due to death or withdrawal from the study. Left truncation applies since only patients that developed AIDS within the study period were recruited, thus only patients with incubation time not exceeding the lapse time were observed; and **Abortion** [26]: contains the entry time and the spontaneous abortion time for 1186 women (197 control group and 989 treatment group exposed to Coumarin derivatives). A censoring proportion of 0.906 occurs due to live birth or induced abortions. Delayed entry to the study is substantial in this dataset: 50% of the control cohort entered the study in week 9 or later, while in the treatment group this occurs for 25% of the cohort. **Implementation:** For our test we used both Gaussian kernels KQIC_Gauss and IMQ kernels KQIC_IMQ. For competing approaches, the implementation is as discussed at the beginning of this section. **Results**: For the Channing house dataset, in Table 2, we observe that all tests agree in not rejecting the null hypothesis for the combined and female groups at a level $\alpha = 0.05$. For the male group, all tests but MinP2 and M&B reject the null hypothesis at $\alpha = 0,05$. Our results agree with [7]. For the AIDS dataset, all tests reach a consensus of rejecting the null, which is consistent with [7], except for MinP2 marked in blue. For the abortion dataset, our test rejects the null hypothesis, suggesting dependency between the entry time $X$ and the spontaneous abortion time $Y$ in both the treatment group and the combined case (in red). This finding is in accordance with domain knowledge [26], where the presence of this dependence was indicated to be due to the study design. The competing tests were unable to detect the dependence; however, did not reject the null hypothesis.

## 6   Conclusions

We address the problem of testing for quasi-independence in the presence of left-truncation, as occurs in real-world examples where events are ordered. The test is nonparametric and general-purpose, can detect a broad class of departures from the null, and applies even where right-censoring is present. In experiments on challenging synthetic data, our method strongly outperforms the alternatives. On real-life datasets, our method yields consistent results to classical approaches where these apply; however, it also detects quasi-dependence in a case where competing general-purpose approaches fail, and where models based on domain knowledge were needed in establishing the result. Our tests are a first step towards the wider challenge of testing quasi-independence in the presence of general physical or causal constraints on the variables, which themselves induces a "baseline" level of dependence. Many real-world settings, apart from left-truncation, do not have the advantage of a "pure" null scenario of perfect independence, and tests must be designed in light of these constraints. These directions are an exciting topic for future study.

# 7    Broader Impact

**Potential benefits to society**    Finding dependencies is a key tool in a broad variety of scientific domains, including clinical treatments, demography, business strategy development, and public policy formulation, with applications spanning the natural and social sciences. Our work addresses these questions by studying the dependence relationships between observed data, where the data already have an intrinsic dependence due to natural order. Moreover, the dependence need not be monotonic, but can take a variety of forms. Detecting the dependence of variables in this setting, which corresponds to many real-life scenarios, will allow scientists or policymakers to better understand their data and research problems, and guide the better design of future research questions. The dependence detection strategy may be used to detect bias in data collection procedures. Such bias could be avoided by verifying the absence of inadvertent dependency relationships in collected data.

**Potential risks to society**    There are a number of ways in which statistical tests can be mis-applied in the wider scientific community, and these must be guarded against. As one example, p-value hacking/failure to correct for multiple testing can result in false positives. In the event that these false positives are surprising or controversial, they can gain considerable traction in the media. In some cases, peoples' health can be at risk. A second risk, specific to tests of dependence, is for correlation and causation to be confused. Our tests detect correlation, however, a misunderstanding of such tests might result in false conclusions of cause and effect. There have been especially pernicious instances when using statistics in domains such as crime prediction.

## Acknowledgement

TF, WX, and AG thank the Gatsby Charitable Foundation for the financial support. MD gratefully acknowledge support from the Deutsche Forschungsgemeinschaft (grant no. PA-2409 5-1).

## Footnotes

[1]Our test-statistic recovers, as a particular case, the squared of this log-rank test by choosing $\mathfrak{K} = 1$

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
