[Supplementary Material]

## Appendix: A kernel test for quasi-independence

## A    Preliminary results

The following Proposition is an intermediate result, which is need to prove Lemmas C.3 and D.1.

**Proposition A.1.** *Define*

$$A_n = \frac{1}{n} \sum_{i=1}^{n} \Delta_i \mathbb{1}_{\{X_i < T_i\}} (\widehat{\pi}^c(X_i, T_i) - \pi^c(X_i, T_i))^2. \tag{9}$$

*Then, the following results hold: i) $A_n \to 0$ almost surely as $n$ grows to infinity and ii) $|A_n| \leq c$, for some constant c, for all large n.*

**Proof:** Since $\widehat{\pi}^c$ and $\pi^c$ are both bounded by 1, we have $|A_n| = A_n \leq 4$ for all $n$ and, thus, ii) is proven.

Let us consider the statement i). It is easy to see that $\mathbb{E}(\mathbb{1}_{\{X_m \leq x, T_m \geq t\}}) = \pi^c(x, t)$. In particular, we have $\mathbb{E}(g(m, i)|X_i, T_i, \Delta_i) = 0$ for $i \neq m$, where $g(m, i) = \mathbb{1}_{\{X_m \leq X_i, T_m \geq T_i\}} - \pi^c(X_i, T_i)$. Now, notice that we can rewrite $A_n$ as $V$-statistic of order 3:

$$A_n = \frac{1}{n} \sum_{i=1}^{n} \Delta_i \mathbb{1}_{\{X_i \leq T_i\}} \left( \frac{1}{n} \sum_{m=1}^{n} (\mathbb{1}_{\{X_m \leq X_i, T_m \geq T_i\}} - \pi^c(X_i, T_i)) \right)^2$$

$$= \frac{1}{n^3} \sum_{i=1}^{n} \sum_{m=1}^{n} \sum_{k=1}^{n} \Delta_i \mathbb{1}_{\{X_i \leq T_i\}} g(m, i) g(k, i).$$

Combining this and the law of large numbers for $V$-statistics yields

$$A_n \overset{a.s.}{\to} \mathbb{E}(\Delta_1 g(1, 2) g(1, 3)) = \mathbb{E}(\Delta_1 \mathbb{E}(g(2, 1) g(3, 1)|X_1, T_1, \Delta_1))$$
$$(\text{independence}) = \mathbb{E}\left(\Delta_1 \mathbb{E}(g(2, 1)|X_1, T_1, \Delta_1) \mathbb{E}(g(3, 1)|X_1, T_1, \Delta_1)\right)$$
$$= 0.$$

∎

## B    Proofs of sections 2 and 3

### B.1    Proof of Proposition 2.1

**Proof:** From Equation (4), we have

$$\Psi_n = \sup_{\omega \in B(\mathcal{H})} \frac{1}{n} \sum_{i=1}^{n} \left( \omega(X_i, Y_i) \widehat{\pi}(X_i, Y_i) - \sum_{k=1}^{n} \omega(X_i, Y_k) \boldsymbol{A}_{ik} \right),$$

where $\boldsymbol{A}_{ik} = \mathbb{1}_{\{X_k \leq X_i < Y_k \leq Y_i\}}/n$.

The previous result and the reproducing kernel property yield

$$\Psi_n^2 = \sup_{\omega \in B_1(\mathcal{H})} \left( \frac{1}{n} \sum_{i=1}^{n} \left( \omega(X_i, Y_i)\widehat{\pi}(X_i, Y_i) - \sum_{k=1}^{n} \omega(X_i, Y_k)\boldsymbol{A}_{ik} \right) \right)^2$$

$$= \sup_{\omega \in B_1(\mathcal{H})} \left\langle \omega(\cdot), \frac{1}{n} \sum_{i=1}^{n} K(X_i, \cdot) \left( L(Y_i, \cdot)\widehat{\boldsymbol{\pi}}_{ii} - \sum_{k=1}^{n} L(Y_k, \cdot)\boldsymbol{A}_{ik} \right) \right\rangle^2$$

$$= \left\| \frac{1}{n} \sum_{i=1}^{n} K(X_i, \cdot) \left( L(Y_i, \cdot)\widehat{\boldsymbol{\pi}}_{ii} - \sum_{k=1}^{n} L(Y_k, \cdot)\boldsymbol{A}_{ik} \right) \right\|_{\mathcal{H}}^2$$

$$= \frac{1}{n^2} \sum_{i,j=1}^{n} \boldsymbol{K}_{ij}\boldsymbol{L}_{ij}\widehat{\boldsymbol{\pi}}_{ii}\widehat{\boldsymbol{\pi}}_{jj} - \frac{2}{n^2} \sum_{i,j,l=1}^{n} \boldsymbol{K}_{ij}\boldsymbol{L}_{il}\widehat{\boldsymbol{\pi}}_{ii}\boldsymbol{A}_{jl} + \frac{1}{n^2} \sum_{i,j,k,l=1}^{n} \boldsymbol{K}_{ij}\boldsymbol{L}_{kl}\boldsymbol{A}_{ik}\boldsymbol{A}_{jl}$$

$$= \frac{1}{n^2} \mathrm{trace}(\boldsymbol{K}\widehat{\boldsymbol{\pi}}\boldsymbol{L}\widehat{\boldsymbol{\pi}} - 2\boldsymbol{K}\widehat{\boldsymbol{\pi}}\boldsymbol{L}\boldsymbol{A}^\mathsf{T} + \boldsymbol{K}\boldsymbol{A}\boldsymbol{L}\boldsymbol{A}^\mathsf{T}),$$

where the second to last equality follows from

$$\frac{1}{n^2} \sum_{i=1}^{n}\sum_{j=1}^{n} \boldsymbol{K}_{ij}\boldsymbol{L}_{ij}\widehat{\boldsymbol{\pi}}_{ii}\widehat{\boldsymbol{\pi}}_{jj} = \frac{1}{n^2} \sum_{i=1}^{n}\sum_{j=1}^{n} \boldsymbol{K}_{ij}(\widehat{\boldsymbol{\pi}}\boldsymbol{L}\widehat{\boldsymbol{\pi}})_{ij} = \frac{1}{n^2}\mathrm{trace}(\boldsymbol{K}\widehat{\boldsymbol{\pi}}\boldsymbol{L}\widehat{\boldsymbol{\pi}}),$$

$$\frac{2}{n^2} \sum_{i,j,l=1}^{n} \boldsymbol{K}_{ij}\boldsymbol{L}_{il}\widehat{\boldsymbol{\pi}}_{ii}\boldsymbol{A}_{jl} = \frac{2}{n^2} \sum_{j=1}^{n}\sum_{l=1}^{n} \left( \sum_{i=1}^{n} \boldsymbol{K}_{ij}(\widehat{\boldsymbol{\pi}}\boldsymbol{L})_{il} \right) \boldsymbol{A}_{jl}$$

$$= \frac{2}{n^2} \sum_{j=1}^{n}\sum_{l=1}^{n} (\boldsymbol{K}\widehat{\boldsymbol{\pi}}\boldsymbol{L})_{jl} \boldsymbol{A}_{lj}^\mathsf{T}$$

$$= \frac{2}{n^2}\mathrm{trace}(\boldsymbol{K}\widehat{\boldsymbol{\pi}}\boldsymbol{L}\boldsymbol{A}^\mathsf{T}),$$

and

$$\frac{1}{n^2} \sum_{i,j,k,l=1}^{n} \boldsymbol{K}_{ij}\boldsymbol{L}_{kl}\boldsymbol{A}_{ik}\boldsymbol{A}_{jl} = \frac{1}{n^2} \sum_{k=1}^{n}\sum_{j=1}^{n} \left( \sum_{i=1}^{n} \boldsymbol{K}_{ij}\boldsymbol{A}_{ik} \right) \left( \sum_{l=1}^{n} \boldsymbol{L}_{kl}\boldsymbol{A}_{jl} \right)$$

$$= \frac{1}{n^2} \sum_{k=1}^{n}\sum_{j=1}^{n} (\boldsymbol{K}\boldsymbol{A})_{jk} (\boldsymbol{L}\boldsymbol{A}^\mathsf{T})_{kj}$$

$$= \frac{1}{n^2}\mathrm{trace}(\boldsymbol{K}\boldsymbol{A}\boldsymbol{L}\boldsymbol{A}^\mathsf{T}). \qquad \blacksquare$$

## B.2 Proof of Proposition 3.3

**Proof:** Equation (6) yields

$$\Psi_{c,n} = \sup_{\omega \in B_1(\mathcal{H})} \frac{1}{n} \sum_{i=1}^{n} \left( \Delta_i \omega(X_i, T_i)\widehat{\pi}^c(X_i, T_i) - \frac{1}{n} \sum_{k=1}^{n} \Delta_k \omega(X_i, T_k)\mathbb{1}_{\{X_k \leq X_i \leq T_k \leq T_i\}} \right)$$

$$= \sup_{\omega \in B_1(\mathcal{H})} \frac{1}{n} \sum_{i=1}^{n} \left( \Delta_i \omega(X_i, T_i)\widehat{\boldsymbol{\pi}}_{ii}^c - \sum_{k=1}^{n} \Delta_k \omega(X_i, T_k)\boldsymbol{B}_{ik} \right),$$

where $\boldsymbol{B}_{ik} = \mathbb{1}_{\{X_k \leq X_i < T_k \leq T_i\}}/n$ and $\widehat{\boldsymbol{\pi}}^c$ is a diagonal matrix with entries $\widehat{\boldsymbol{\pi}}_{ii}^c = \widehat{\pi}^c(X_i, T_i)$.

Then, by following the exact same computations of the proof of Proposition 2.1, we deduce

$$\Psi_{c,n}^2 = \frac{1}{n^2}\mathrm{trace}(\boldsymbol{K}\widehat{\boldsymbol{\pi}}\tilde{\boldsymbol{L}}\widehat{\boldsymbol{\pi}} - 2\boldsymbol{K}\widehat{\boldsymbol{\pi}}\tilde{\boldsymbol{L}}\boldsymbol{B}^\mathsf{T} + \boldsymbol{K}\boldsymbol{B}\tilde{\boldsymbol{L}}\boldsymbol{B}^\mathsf{T}),$$

where $\tilde{\boldsymbol{L}}_{ik} = \Delta_i \Delta_k L(T_i, T_k)$. $\qquad \blacksquare$

## B.3 Proof of Proposition 3.2

**Proof:** Under Assumption 3.1, we have that for all $x \leq y$,

$$\pi_1^c(x, y) = \mathbb{P}(X \leq x, T \geq y, \Delta = 1) = \mathbb{E}\left(\mathbb{1}_{\{X \leq x, Y \geq y\}} \mathbb{E}\left(\mathbb{1}_{\{C \geq Y\}} | X, Y\right)\right)$$
$$= \mathbb{E}\left(\mathbb{1}_{\{X \leq x, Y \geq y\}} S_{C|X}(Y)\right)$$
$$= \int_0^x \int_y^\infty S_{C|X=x'}(y') f_{XY}(x', y') dx', dy',$$

and

$$\pi^c(x, y) = \mathbb{P}(X \leq x, T \geq y) = \mathbb{E}\left(\mathbb{1}_{\{X \leq x\}} S_{C|X}(y) S_{Y|X}(y)\right)$$
$$= \int_0^x S_{C|X=x'}(y) S_{Y|X=x'}(y) f_X(x') dx'.$$

The null hypothesis states $f_{XY}(x, y) = \tilde{f}_X(x) \tilde{f}_Y(y)$ for all $x \leq y$ such that $S_{T|X=x}(y) > 0$. Thus

$$\pi_1^c(x, y) = \int_0^x \int_y^\infty S_{C|X=x'}(y') \tilde{f}_X(x') \tilde{f}_Y(y') dx' dy',$$

$$\pi^c(x, y) = \tilde{S}_Y(y) \int_0^x S_{C|X=x'}(y) \tilde{f}_X(x') dx'.$$

By using the previous result, it is easy to see that, under the null,

$$-\pi^c(x, y) \frac{\partial^2}{\partial x \partial y} \pi_1^c(x, y) = \left(\tilde{S}_Y(y) \int_0^x S_{C|X=x'}(y) \tilde{f}_X(x') dx'\right) S_{C|X=x}(y) \tilde{f}_X(x) \tilde{f}_Y(y),$$

and

$$\frac{\partial \pi^c(x, y)}{\partial x} \frac{\partial \pi_1^c(x, y)}{\partial y} = -\left(\tilde{S}_Y(y) S_{C|X=x}(y) \tilde{f}_X(x)\right) \int_0^x S_{C|X=x'}(y) \tilde{f}_X(x') dx' \tilde{f}_Y(y)$$
$$= -\left(\tilde{S}_Y(y) \int_0^x S_{C|X=x'}(y) \tilde{f}_X(x') dx'\right) S_{C|X=x}(y) \tilde{f}_X(x) \tilde{f}_Y(y),$$

from which it follows that $\rho^c = 0$, and thus $\Psi = 0$. ∎

## C  Proof of Theorem 4.1

Before proving Theorem 4.1 we give some essential definitions which will be used by our proofs. We will first introduce Lemma C.1, which is an essential step in the proof of Theorem 4.1. A full proof for Lemma C.1 is given later in this section.

Our data are considered to live in a common filtrated probability space $(\Omega, \mathcal{F}, (\mathcal{F}_t)_{t \geq 0}, \mathbb{P})$, where $\mathcal{F}$ is the natural $\sigma$-algebra, and $\mathcal{F}_t$ is the filtration generated by

$$\left\{\mathbb{1}_{\{T_i \leq s, \Delta_i = 1\}}, \mathbb{1}_{\{T_i \leq s, \Delta_i = 0\}}, X_i : 0 \leq s \leq t, i \in [n]\right\},$$

and the $\mathbb{P}$-null sets of $\mathcal{F}$.

We define $\tau_n = \max\{T_1, \ldots, T_n\}$. For each $i \in [n]$, we define the $i$-th individual counting and risk processes, $N_i(t)$ and $Y_i(t)$, by $N_i(t) = \Delta_i \mathbb{1}_{\{T_i \leq t\}}$ and $Y_i(t) = \mathbb{1}_{\{T_i \geq t\}}$, respectively. For each individual $i$, we define the process $(M_i(t))_{t \geq 0}$ by

$$M_i(t) = N_i(t) - \int_{(0,t]} \mathbb{1}_{\{X_i \leq s\}} Y_i(s) \tilde{\lambda}_Y(s) ds.$$

It is standard to verify that $M_i(t)$ is an $(\mathcal{F}_t)$-martingale under the null hypothesis, and that, for any bounded predictable process $(H_i(t))_{t \geq 0}$, $\int_{(0,t]} H_i(s) dM_i(s)$ is also an $(\mathcal{F}_t)$-martingale under the null hypothesis.

Let $(T_1', \Delta_1', X_1')$ and $(T_2', \Delta_2', X_2')$ be independent copies of our data $((T_i, \Delta_i, X_i))_{i=1}^n$. Sometimes our results are written in terms of $\tilde{\mathbb{E}}$ which is defined by $\tilde{\mathbb{E}}(\cdot) = \mathbb{E}\left(\cdot | ((T_i, \Delta_i, X_i))_{i=1}^n\right)$. Additionally,

we denote by $Y_1'$ and $Y_2'$, the individual risk functions associated to $T_1'$ and $T_2'$, which are defined by $Y_1'(t) = \mathbb{1}_{\{T_1' \geq t\}}$ and $Y_2'(t) = \mathbb{1}_{\{T_2' \geq t\}}$, respectively. Finally, we define $Z_i(t) = \omega(X_i, t)\mathbb{1}_{\{X_i \leq t\}}$ for all $i \in [n]$, and, based on $(T_1', \Delta_1', X_1')$ and $(T_2', \Delta_2', X_2')$, we define $Z_1'(t) = \omega(X_1', t)\mathbb{1}_{\{X_1' \leq t\}}$ and $Z_2'(t) = \omega(X_2', t)\mathbb{1}_{\{X_2' \leq t\}}$.

**Lemma C.1.** *Assume that $\mathfrak{K}$ is bounded. Then, under the null hypothesis*

$$\sqrt{n}\Psi_{n,c} = \sup_{\omega \in B_1(\mathcal{H})} \frac{1}{\sqrt{n}} \sum_{i=1}^{n} \int_0^{\tau_n} \left( Z_i(t)\pi^c(X_i, t) - \tilde{\mathbb{E}}\left( Z_1'(t)Y_1'(t)\mathbb{1}_{\{X_i \leq X_1'\}} \right) \right) dM_i(t) + o_p(1).$$

### C.1 Proof of Theorem 4.1

By Lemma C.1,

$$\sqrt{n}\Psi_{n,c} = \sup_{\omega \in B_1(\mathcal{H})} \frac{1}{\sqrt{n}} \sum_{i=1}^{n} \int_0^{\tau_n} \left( Z_i(t)\pi^c(X_i, t) - \tilde{\mathbb{E}}\left( Z_1'(t)Y_1'(t)\mathbb{1}_{\{X_i \leq X_1'\}} \right) \right) dM_i(t) + o_p(1).$$

Observe that, by the reproducing kernel property, we have $Z_i(t) = \langle \omega, \mathfrak{K}((X_i, t), \cdot) \rangle_{\mathcal{H}} \mathbb{1}_{\{X_i \leq t\}}$ and $Z_1'(t) = \langle \omega, \mathfrak{K}((X_1', t), \cdot) \rangle_{\mathcal{H}} \mathbb{1}_{\{X_1' \leq t\}}$. Thus,

$$\left( Z_i(t)\pi^c(X_i, t) - \tilde{\mathbb{E}}\left( Z_1'(t)Y_1'(t)\mathbb{1}_{\{X_i \leq X_1'\}} \right) \right)$$

$$= \left( \langle \omega, \mathfrak{K}((X_i, t), \cdot) \rangle_{\mathcal{H}} \mathbb{1}_{\{X_i \leq t\}}\pi^c(X_i, t) - \tilde{\mathbb{E}}\left( \langle \omega, \mathfrak{K}((X_1', t), \cdot) \rangle_{\mathcal{H}} \mathbb{1}_{\{X_1' \leq t\}}Y_1'(t)\mathbb{1}_{\{X_i \leq X_1'\}} \right) \right)$$

$$= \left\langle \omega, \mathfrak{K}((X_i, t), \cdot)\mathbb{1}_{\{X_i \leq t\}}\pi^c(X_i, t) - \tilde{\mathbb{E}}\left( \mathfrak{K}((X_1', t), \cdot)\mathbb{1}_{\{X_1' \leq t\}}Y_1'(t)\mathbb{1}_{\{X_i \leq X_1'\}} \right) \right\rangle_{\mathcal{H}},$$

where the second equality follows from the linearity of expectation, assuming Bochner integrability of the feature map (true for bounded $\mathfrak{K}$). To ease notation, we define the functions $a : \mathbb{R}^2 \to \mathbb{R}$ and $b : \mathbb{R}^3 \to \mathbb{R}$ by $a(X_i, t) = \mathbb{1}_{\{X_i \leq t\}}\pi^c(X_i, t)$ and $b(X_1', X_i, t) = Y_1'(t)\mathbb{1}_{\{X_i \leq X_1' \leq t\}}$, respectively, and write

$$\left( Z_i(t)\pi^c(X_i, t) - \tilde{\mathbb{E}}\left( Z_1'(t)Y_1'(t)\mathbb{1}_{\{X_i \leq X_1'\}} \right) \right)$$

$$= \left\langle \omega, \mathfrak{K}((X_i, t), \cdot)a(X_i, t) - \tilde{\mathbb{E}}\left( \mathfrak{K}((X_1', t), \cdot)b(X_1', X_i, t) \right) \right\rangle_{\mathcal{H}}. \tag{10}$$

From the previous result, it is easy to see that

$$\frac{1}{\sqrt{n}} \sum_{i=1}^{n} \int_0^{\tau_n} \left( Z_i(t)\pi^c(X_i, t) - \tilde{\mathbb{E}}\left( Z_1'(t)Y_1'(t)\mathbb{1}_{\{X_i \leq X_1'\}} \right) \right) dM_i(t)$$

$$= \left\langle \omega, \frac{1}{\sqrt{n}} \sum_{i=1}^{n} \int_0^{\tau_n} \left( \mathfrak{K}((X_i, t), \cdot)a(X_i, t) - \tilde{\mathbb{E}}\left( \mathfrak{K}((X_1', t), \cdot)b(X_1', X_i, t) \right) \right) dM_i(t) \right\rangle_{\mathcal{H}},$$

and thus

$$\sup_{\omega \in B_1(\mathcal{H})} \left( \frac{1}{\sqrt{n}} \sum_{i=1}^{n} \int_0^{\tau_n} \left( Z_i(t)\pi^c(X_i, t) - \tilde{\mathbb{E}}\left( Z_1'(t)Y_1'(t)\mathbb{1}_{\{X_i \leq X_1'\}} \right) \right) dM_i(t) \right)^2$$

$$\left\| \frac{1}{\sqrt{n}} \sum_{i=1}^{n} \int_0^{\tau_n} \left( \mathfrak{K}((X_i, t), \cdot)a(X_i, t) - \tilde{\mathbb{E}}\left( \mathfrak{K}((X_1', t), \cdot)b(X_1', X_i, t) \right) \right) dM_i(t) \right\|_{\mathcal{H}}^2$$

$$= \frac{1}{n} \sum_{i=1}^{n} \sum_{j=1}^{n} J((T_i, \Delta_i, X_i), (T_j, \Delta_j, X_j)), \tag{11}$$

where the function $J : (\mathbb{R} \times \{0, 1\} \times \mathbb{R})^2 \to \mathbb{R}$ is defined by

$$J((s, r, x), (s', r', x')) = \int_0^s \int_0^{s'} A((t, x), (t', x'))dm_{s',r',x'}(t')dm_{s,r,x}(t),$$

$dm_{s,r,x}(t) = r\delta_s(t) - \mathbb{1}_{\{s\geq t\}}\mathbb{1}_{\{x\leq t\}}\tilde{\lambda}_Y(t)dt$ (notice that $dM_i(t) = dm_{T_i,\Delta_i,X_i}(t)$), and $A : (\mathbb{R} \times \mathbb{R})^2 \to \mathbb{R}$ is defined as

$$A((t,x),(t',x'))$$
$$= \Big\langle \mathfrak{K}((x,t),\cdot)a(x,t) - \tilde{\mathbb{E}}\left(\mathfrak{K}((X_1',t),\cdot)b(X_1',x,t)\right)$$
$$, \mathfrak{K}((x',t'),\cdot)a(x',t') - \tilde{\mathbb{E}}\left(\mathfrak{K}((X_2',t'),\cdot)b(X_2',x',t')\right)\Big\rangle_{\mathcal{H}}$$
$$= \mathfrak{K}((x,t),(x',t'))a(x,t)a(x',t') - \tilde{\mathbb{E}}(\mathfrak{K}((X_1',t),(x',t'))b(X_1',x,t)a(x',t'))$$
$$- \tilde{\mathbb{E}}(\mathfrak{K}((x,t),(X_2',t'))a(x,t)b(X_2',x',t')) + \tilde{\mathbb{E}}(\mathfrak{K}((X_1',t),(X_2',t'))b(X_1',x,t)b(X_2',x',t')).$$

It is not difficult to verify that the sum in Equation (11) is a degenerate $V$-statistic. Indeed, the degeneracy property can be verified by noticing that

$$\mathbb{E}(J((T_i,\Delta_i,X_i),(s',r',x')))$$
$$= \mathbb{E}\left(\int_0^{T_i}\left(\int_0^{s'}A((t,X_i),(t',x'))dm_{s',r',x}(t')\right)dM_i(t)\right)$$
$$= \mathbb{E}(Q(T_i)),$$

where $Q(s) = \int_0^s \left(\int_0^{s'} A((t,X_i),(t',x'))dm_{s',r',x}(t')\right)dM_i(t)$ is a zero mean $(\mathcal{F}_s)$-martingale, and thus, by the optional stopping Theorem, $\mathbb{E}(Q(T_i)) = \mathbb{E}(Q(0)) = 0$. Then, by [23, Theorem 4.3.2], we deduce

$$\frac{1}{n}\sum_{i=1}^n\sum_{j=1}^n J((T_i,\Delta_i,X_i),(T_j,\Delta_j,X_j)) \xrightarrow{\mathcal{D}} \mathbb{E}(J((T_1,\Delta_1,X_1),(T_1,\Delta_1,X_1))) + \mathcal{Y},$$

where $\mathcal{Y} = \sum_{i=1}^\infty \lambda_i(\xi_i^2 - 1)$, $\xi_1,\xi_2,\dots$ are independent standard normal random variables, and $\lambda_1,\lambda_2,\dots$ are positive constants.

The previous result, together with Lemma C.1, allow us to deduce

$$\Psi_{c,n}^2 \xrightarrow{\mathcal{D}} \mu + \mathcal{Y},$$

where $\mu = \mathbb{E}(J((T_1,\Delta_1,X_1),(T_1,\Delta_1,X_1)))$. Notice that all integrability conditions are satisfied as we assume the reproducing kernel is bounded.

## C.2 Proof of Lemma C.1

In order to prove Lemma of C.1, we require some intermediate results.

Recall that our test-statistic is computed as the supremum over $\omega \in B_1(\mathcal{H})$ of sums

$$\frac{1}{n}\sum_{i=1}^n \Delta_i\omega(X_i,T_i)\hat{\pi}^c(X_i,T_i) - \frac{1}{n^2}\sum_{i=1}^n\sum_{k=1}^n \Delta_k\omega(X_i,T_k)\mathbb{1}_{\{X_k\leq X_i < T_k \leq T_i\}}.$$

By using the notation introduced at the beginning of Section C, the previous sum can be rewritten as

$$\frac{1}{n}\sum_{i=1}^n\left(\Delta_i\omega(X_i,T_i)\hat{\pi}^c(X_i,T_i) - \frac{1}{n}\sum_{k=1}^n \Delta_i\omega(X_k,T_i)\mathbb{1}_{\{X_i\leq X_k < T_i \leq T_k\}}\right)$$
$$= \frac{1}{n}\sum_{i=1}^n\int_0^{T_i}\left(\omega(X_i,y)\mathbb{1}_{\{X_i\leq y\}}\hat{\pi}^c(X_i,y) - \frac{1}{n}\sum_{k=1}^n \omega(X_k,y)\mathbb{1}_{\{X_k\leq y\}}\mathbb{1}_{\{y\leq T_k\}}\mathbb{1}_{\{X_i\leq X_k\}}\right)dN_i(y)$$
$$= \frac{1}{n}\sum_{i=1}^n\int_0^{T_i}\left(Z_i(y)\hat{\pi}^c(X_i,y) - \frac{1}{n}\sum_{k=1}^n Z_k(y)Y_k(y)\mathbb{1}_{\{X_i\leq X_k\}}\right)dN_i(y)$$
$$= \frac{1}{n}\sum_{i=1}^n\int_0^{T_i} H_i(y)dN_i(y),$$

where $H_i(y) = Z_i(y)\widehat{\pi}^c(X_i, y) - \frac{1}{n}\sum_{k=1}^{n} Z_k(y)Y_k(y)\mathbb{1}_{\{X_i \leq X_k\}}$. Thus,

$$\Psi_{n,c} = \sup_{\omega \in B_1(\mathcal{H})} \frac{1}{n} \sum_{i=1}^{n} \int_0^{\tau_n} H_i(y)dN_i(y), \qquad (12)$$

where recall that $\tau_n = \max\{T_1, \ldots, T_n\}$.

**Proposition C.2.** *Assume that $\mathfrak{K}$ is bounded. Then, under the null hypothesis, the process $(W(t))_{t \geq 0}$, defined by $W(t) = \frac{1}{n}\sum_{i=1}^{n}\int_0^t H_i(y)dN_i(y)$, is an $(\mathcal{F}_t)$-martingale, and can be rewritten as*

$$W(t) = \frac{1}{n}\sum_{i=1}^{n} \int_0^t H_i(y)dM_i(y).$$

Notice that the previous proposition, and Equation (12) suggest the result of Lemma C.1. It remains to prove that the process $H_i(y)$ may be approximated by its "population limit". We prove this result in two steps in the two lemmas below.

**Lemma C.3.** *Assume that $\mathfrak{K}$ is bounded. Then, under the null hypothesis*

$$\sup_{\omega \in B_1(\mathcal{H})} \frac{1}{\sqrt{n}} \sum_{i=1}^{n} \int_0^{\tau_n} Z_i(y)\left(\widehat{\pi}^c(X_i, y) - \pi^c(X_i, y)\right)dM_i(y) = o_p(1),$$

**Lemma C.4.** *Assume that $\mathfrak{K}$ is bounded. Then, under the null hypothesis*

$$\sup_{\omega \in B_1(\mathcal{H})} \frac{1}{\sqrt{n}} \sum_{i=1}^{n} \int_0^{\tau_n} \left(\frac{1}{n}\sum_{j=1}^{n} Z_j(y)Y_j(y)\mathbb{1}_{\{X_i \leq X_j\}} - \tilde{\mathbb{E}}(Z_1'(y)Y_1'(y)\mathbb{1}_{\{X_i \leq X_1'\}})\right)dM_i(y) = o_p(1),$$

**Proof of Lemma C.1:** Equation (12) and Lemma C.2 yield

$$\sqrt{n}\Psi_{n,c} = \sup_{\omega \in B_1(\mathcal{H})} \frac{1}{\sqrt{n}} \sum_{i=1}^{n} \int_0^{\tau_n} H_i(y)dM_i(y),$$

where (recall) $H_i(y) = Z_i(y)\widehat{\pi}^c(X_i, y) - \frac{1}{n}\sum_{k=1}^{n} Z_k(y)Y_k(y)\mathbb{1}_{\{X_i \leq X_k\}}$. Notice that to obtain the result, we need to replace $\widehat{\pi}^c$ by its population version $\pi^c$, and, given $(T_i, \Delta_i, X_i)$, we need to replace the i.i.d. sum $\frac{1}{n}\sum_{k=1}^{n} Z_k(y)Y_k(y)\mathbb{1}_{\{X_i \leq X_k\}}$ by its limit, which is given by $\tilde{\mathbb{E}}(Z_1'(y)Y_1'(y)\mathbb{1}_{\{X_i \leq X_1'\}})$. By the triangular inequality, this result follows from lemmas C.3 and C.4.

## C.3 Proofs of Proposition C.2, and Lemmas C.3 and C.4

### C.3.1 Proof of Proposition C.2

Recall that $dM_i(y) = dN_i(y) - \mathbb{1}_{\{X_i \leq y\}}Y_i(y)\tilde{\lambda}_Y(y)dy$. A straightforward computation verifies $\frac{1}{n}\sum_{i=1}^{n}\int_0^t H_i(y)\mathbb{1}_{\{X_i \leq y\}}Y_i(y)\tilde{\lambda}_Y(y)dy = 0$ for all $t \geq 0$, and thus

$$W(t) = \frac{1}{n}\sum_{i=1}^{n} \int_0^t H_i(y)dM_i(y).$$

Also, notice that $(H_i(t))_{t \geq 0}$ (with $\omega \in B_1(\mathcal{H})$) is bounded and $(\mathcal{F}_t)$-predictable, and that $M_i(t)$ is an $(\mathcal{F}_t)-$ martingale under the null hypothesis. Then, by standard martingale results we deduce that $(W(t))_{t \geq 0}$ is an $(\mathcal{F}_t)$-martingale.

### C.3.2 Proof of Lemma C.3

Observe that

$$Z_i(t)(\widehat{\pi}^c(X_i, t) - \pi^c(X_i, t)) = \langle\omega, \mathfrak{K}((X_i, t), \cdot)\rangle_{\mathcal{H}} \mathbb{1}_{\{X_i \leq t\}}(\widehat{\pi}^c(X_i, t) - \pi^c(X_i, t))$$

since $Z_i(t, \omega) = \omega(X_i, t)\mathbb{1}_{\{X_i \leq t\}} = \langle\omega, \mathfrak{K}((X_i, t), \cdot)\rangle_{\mathcal{H}}\mathbb{1}_{\{X_i \leq t\}}$ due to the reproducing property.

Then,

$$\sup_{\omega \in B_1(\mathcal{H})} \left( \frac{1}{\sqrt{n}} \sum_{i=1}^{n} \int_0^{\tau_n} Z_i(t) \left( \widehat{\pi}^c(X_i, t) - \pi^c(X_i, t) \right) dM_i(t) \right)^2$$

$$= \sup_{\omega \in B_1(\mathcal{H})} \left( \frac{1}{\sqrt{n}} \sum_{i=1}^{n} \int_0^{\tau_n} \langle \omega, \mathfrak{K}((X_i, t), \cdot) \rangle_{\mathcal{H}} \mathbb{1}_{\{X_i \leq t\}} (\widehat{\pi}^c(X_i, t) - \pi^c(X_i, t)) dM_i(t) \right)^2$$

$$= \sup_{\omega \in B_1(\mathcal{H})} \left\langle \omega, \frac{1}{\sqrt{n}} \sum_{i=1}^{n} \int_0^{\tau_n} \mathfrak{K}((X_i, t), \cdot) \mathbb{1}_{\{X_i \leq t\}} (\widehat{\pi}^c(X_i, t) - \pi^c(X_i, t)) dM_i(t) \right\rangle_{\mathcal{H}}^2$$

$$= \frac{1}{n} \sum_{i=1}^{n} \sum_{k=1}^{n} \int_0^{\tau_n} \int_0^{\tau_n} J((X_i, t), (X_k, s)) dM_i(t) dM_k(s),$$

where

$$J((X_i, t), (X_k, s))$$
$$= \mathfrak{K}((X_i, t), (X_k, s)) \mathbb{1}_{\{X_i \leq t\}} \mathbb{1}_{\{X_k \leq s\}} (\widehat{\pi}^c(X_i, t) - \pi^c(X_i, t)) (\widehat{\pi}^c(X_k, s) - \pi^c(X_k, s)) \quad (13)$$

Define the process $(Q(y))_{y \geq 0}$ by

$$Q(y) = \frac{1}{n} \sum_{i=1}^{n} \sum_{k=1}^{n} \int_0^y \int_0^y J((X_i, t), (X_k, s)) dM_i(t) dM_k(s),$$

and notice that we wish to prove that $Q(\tau_n) = o_p(1)$. Let $\delta > 0$, then, by Markov's inequality,

$$\mathbb{P}(Q(\tau_n) > \delta) \leq \frac{\mathbb{E}(Q(\tau_n))}{\delta} = \frac{\mathbb{E}(Q_D(\tau_n))}{\delta} + \frac{2\mathbb{E}(Q_{D^c}(\tau_n))}{\delta},$$

where the last equality holds since, by symmetry, $Q(y) = Q_D(y) + 2Q_{D^c}(y)$, where

$$Q_D(y) = \frac{1}{n} \sum_{i=1}^{n} \sum_{k=1}^{n} \int_0^y \int_0^y \mathbb{1}_{\{s=t\}} J((X_i, t), (X_k, s)) dM_i(t) dM_k(s), \quad (14)$$

and

$$Q_{D^c}(y) = \frac{1}{n} \sum_{k=1}^{n} \sum_{i=1}^{n} \int_0^y \int_{(0,s)} J((X_i, t), (X_k, s)) dM_i(t) dM_k(s).$$

By [10, Theorem 6.8], $Q_{D^c}(y)$ is an $(\mathcal{F}_y)$-martingale, and, by the optional stopping theorem, $\mathbb{E}(Q_{D^c}(\tau_n)) = \mathbb{E}(Q_{D^c}(0)) = 0$. Thus

$$\mathbb{P}(Q(\tau_n) > \delta) \leq \frac{\mathbb{E}(Q_D(\tau_n))}{\delta},$$

where

$$Q_D(\tau_n) = \frac{1}{n} \sum_{i=1}^{n} \sum_{k=1}^{n} \int_0^{\tau_n} J((X_i, t), (X_k, t)) d[M_i, M_k](t)$$

$$= \frac{1}{n} \sum_{i=1}^{n} \int_0^{\tau_n} J((X_i, t), (X_i, t)) d[M_i](t)$$

$$= \frac{1}{n} \sum_{i=1}^{n} \int_0^{\tau_n} J((X_i, t), (X_i, t)) N_i(t)$$

$$= \frac{1}{n} \sum_{i=1}^{n} \Delta_i J((X_i, T_i), (X_i, T_i))$$

follows from considering continuous survival and censoring times.

We finish the proof by proving $\mathbb{E}(Q_D(\tau_n)) \to 0$ as $n$ tends to infinity. Observe that

$$\mathbb{E}(Q_D(\tau_n))$$

$$= \mathbb{E}\left(\frac{1}{n}\sum_{i=1}^{n}\Delta_i J((X_i, T_i), (X_i, T_i))\right) \leq c_1 \mathbb{E}\left(\frac{1}{n}\sum_{i=1}^{n}\Delta_i \mathbb{1}_{\{X_i \leq T_i\}}(\widehat{\pi}^c(X_i, T_i) - \pi^c(X_i, T_i))^2\right)$$

follows from substituting the function $J$ with the expression given in Equation (13), and by assuming the reproducing kernel is bounded by some constant $c_1 > 0$. By Proposition A.1, the sum $\frac{1}{n}\sum_{i=1}^{n}\Delta_i\mathbb{1}_{\{X_i \leq T_i\}}(\widehat{\pi}^c(X_i, T_i) - \pi^c(X_i, T_i))^2$ converges to 0 almost surely, and it is bounded by some constant $c > 0$, then the desired result follows from an application of dominated convergence.

### C.3.3 Proof of Lemma C.4

Notice that, by the reproducing property,

$$\frac{1}{n}\sum_{j=1}^{n}Z_j(t)Y_j(t)\mathbb{1}_{\{X_i \leq X_j\}} - \tilde{\mathbb{E}}(Z_1'(t)Y_1'(t)\mathbb{1}_{\{X_i \leq X_1'\}})$$

$$= \frac{1}{n}\sum_{j=1}^{n}\langle\omega, \mathfrak{K}((X_j, t), \cdot)\rangle_{\mathcal{H}}Y_j(t)\mathbb{1}_{\{X_i \leq X_j \leq t\}} - \tilde{\mathbb{E}}\left(\langle\omega, \mathfrak{K}((X_1', t), \cdot)\rangle_{\mathcal{H}}Y_1'(t)\mathbb{1}_{\{X_i \leq X_1' \leq t\}}\right)$$

$$= \left\langle\omega, \frac{1}{n}\sum_{j=1}^{n}\mathfrak{K}((X_j, t), \cdot)Y_j(t)\mathbb{1}_{\{X_i \leq X_j \leq t\}} - \tilde{\mathbb{E}}\left(\mathfrak{K}((X_1', t), \cdot)Y_1'(t)\mathbb{1}_{\{X_i \leq X_1' \leq t\}}\right)\right\rangle_{\mathcal{H}}.$$

To ease notation, we define $a_{ij}(t) = Y_j(t)\mathbb{1}_{\{X_i \leq X_j \leq t\}}$ and $b_{i1}'(t) = Y_1'(t)\mathbb{1}_{\{X_i \leq X_1' \leq t\}}$ (similarly, we define $b_{i2}'(t) = Y_2'(t)\mathbb{1}_{\{X_i \leq X_2' \leq t\}}$, where recall that $(T_1', \Delta_1', X_1')$ and $(T_2', \Delta_2', X_2')$ are independent copies of our data). Then, the previous term can be rewritten as

$$\frac{1}{n}\sum_{j=1}^{n}Z_j(t)Y_j(t)\mathbb{1}_{\{X_i \leq X_j\}} - \tilde{\mathbb{E}}(Z_1'(t)Y_1'(t)\mathbb{1}_{\{X_i \leq X_1'\}})$$

$$= \left\langle\omega, \frac{1}{n}\sum_{j=1}^{n}\mathfrak{K}((X_j, t), \cdot)a_{ij}(t) - \tilde{\mathbb{E}}\left(\mathfrak{K}((X_1', t), \cdot)b_{i1}'(t)\right)\right\rangle_{\mathcal{H}}.$$

By using the fact we take supremum on the unit ball of an RKHS, it is not difficult to deduce,

$$\sup_{\omega \in B_1(\mathcal{H})}\left(\frac{1}{\sqrt{n}}\sum_{i=1}^{n}\int_0^{\tau_n}\left(\frac{1}{n}\sum_{j=1}^{n}Z_j(t)Y_j(t)\mathbb{1}_{\{X_i \leq X_j\}} - \tilde{\mathbb{E}}(Z_1'(t)Y_1'(t)\mathbb{1}_{\{X_i \leq X_1'\}})\right)dM_i(y)\right)^2$$

$$= \frac{1}{n}\sum_{i=1}^{n}\sum_{k=1}^{n}\int_0^{\tau_n}\int_0^{\tau_n}J((X_i, t), (X_k, s))dM_i(t)dM_k(s), \tag{15}$$

where

$$J((X_i, t), (X_k, s))$$

$$= \frac{1}{n^2}\sum_{j=1}^{n}\sum_{l=1}^{n}\mathfrak{K}((X_j, t), (X_l, s))a_{ij}(t)a_{kl}(s) - \frac{1}{n}\sum_{l=1}^{n}\tilde{\mathbb{E}}(\mathfrak{K}((X_1', t), (X_l, s))b_{i1}'(t)a_{kl}(s)$$

$$- \frac{1}{n}\sum_{j=1}^{n}\tilde{\mathbb{E}}(\mathfrak{K}((X_j, t), (X_2', s))a_{ij}(t)b_{k2}'(s) + \tilde{\mathbb{E}}(\mathfrak{K}((X_1', t), (X_2', s))b_{i1}'(t)b_{k2}'(s)), \tag{16}$$

Following the same steps of the proof of Lemma C.3, we can prove that Equation (15) is $o_p(1)$ by proving that

$$\mathbb{E}\left(\frac{1}{n}\sum_{i=1}^{n}J((X_i, T_i), (X_i, T_i))\right) \to 0. \tag{17}$$

For this purpose, first observe that

$$\frac{1}{n}\sum_{i=1}^{n} J((X_i, T_i), (X_i, T_i))$$

$$= \frac{1}{n^3}\sum_{i,j,l=1}^{n} \mathfrak{K}((X_j, T_i), (X_l, T_i)) \, a_{ij}(T_i)a_{il}(T_i) - \frac{2}{n^2}\sum_{i,l=1}^{n} \tilde{\mathbb{E}}(\mathfrak{K}((X'_1, T_i), (X_l, T_i))b'_{i1}(T_i)a_{il}(T_i)$$

$$+ \frac{1}{n}\sum_{i=1}^{n} \tilde{\mathbb{E}}(\mathfrak{K}((X'_1, T_i), (X'_2, T_i))b'_{i1}(T_i)b'_{i2}(T_i).$$

Each sum on the right-hand side of the previous equation is a $V$-statistic of order 3, 2 and 1, respectively. It can easily be seen that they all converge to the same limit. Consequently, the law of large numbers for $V$-statistics implies that

$$\frac{1}{n}\sum_{i=1}^{n} J((X_i, T_i), (X_i, T_i)) \to 0$$

almost surely. Since the reproducing kernel is assumed to be bounded and, thus, the sum is bounded as well, we can deduce, finally, (17) from the dominated convergence theorem.

## D   Proof of Theorem 4.2

The consistency proof relies on the interpretation of the test statistic $\Psi_{c,n}$ and the KQIC $\Psi_c$ as the Hilbert space distances of embeddings of certain positive measures. These distances measure the degree of (quasi)-dependence. In this spirit, this approach is connected to the well-established Hilbert Schmidt Independence Criterion, see e.g. [5, 16, 27, 29].

Now, let us become more concrete and introduce the following measures $\nu_0$ and $\nu_1$ on $R_+^2$ given by

$$\nu_0(dx, dy) = \pi^c(x, y)\pi_1^c(dx, dy)$$
$$= \pi^c(x, y)S_{C|X=x}(y)f_{XY}(x, y)dxdy,$$

$$\nu_1(dx, dy) = \pi^c(dx, dy)\pi_1^c(x, dy)$$
$$= \left(S_{Y|X=x}(y)S_{C|X=x}(y)f_X(x)\right)\left(\int_0^x S_{C|X=t}(y)f_{XY}(t, y)dt\right)dxdy$$

as well as their empirical counterparts $\nu_0^n$ and $\nu_1^n$ defined as

$$\nu_0^n(dx, dy) = \frac{\widehat{\pi}^c(x, y)}{n}\sum_{i=1}^{n} \Delta_i \delta_{X_i}(x)\delta_{T_i}(y)$$

$$\nu_1^n(dx, dy) = \frac{\mathbb{1}_{\{x \le y\}}}{n^2}\left(\sum_{i=1}^{n} \delta_{X_i}(x)\mathbb{1}_{\{T_i \ge y\}}\right)\left(\sum_{k=1}^{n} \Delta_k \delta_{T_k}(y)\mathbb{1}_{\{X_k \le x\}}\right).$$

Moreover, set $\widehat{\rho}^c = \nu_0^n - \nu_1^n$, which is the empirical counterpart of the measure induced by the density $\rho^c$. Then the embeddings of the (empirical) measures into the underlying RKHS are given by

$$\phi_j(\cdot) = \iint_{x \le y} \mathfrak{K}((x, y), \cdot)\nu_j(dx, dy) \quad \text{and} \quad \phi_j^n(\cdot) = \iint_{x \le y} \mathfrak{K}((x, y), \cdot)\nu_j^n(dx, dy).$$

By straightforward calculations, we obtain

$$\Psi_{c,n}^2 = \sup_{\omega \in B_1(\mathcal{H})}\left(\iint_{x \le y} \omega(x, y)\widehat{\rho}^c(dx, dy)\right)^2 = \|\phi_0^n - \phi_1^n\|_{\mathcal{H}}^2$$

and

$$\Psi_c^2 = \sup_{\omega \in B_1(\mathcal{H})}\left(\iint_{x \le y} \omega(x, y)\rho^c(dx, dy)\right)^2 = \|\phi_0 - \phi_1\|_{\mathcal{H}}^2.$$

Consequently, the first part of Theorem 4.2 follows from convergence of the aforementioned distances:

**Lemma D.1.** *We have* $\|\phi_0^n - \phi_1^n\|_{\mathcal{H}}^2 \to \|\phi_0 - \phi_1\|_{\mathcal{H}}^2$ *in probability.*

The proof of Lemma D.1 is given below. For the second part of Theorem 4.2, recall that by assumption the chosen kernel $\mathfrak{K}$ is $c_0$-universal and, thus, the embedding of finite signed Borel measures is injective, see [31] for details. In particular, $\Psi_c^2 = \|\phi_0 - \phi_1\|_{\mathcal{H}}^2$ equals zero if and only if $\nu_0 \equiv \nu_1$, or equivalently $\rho^c(x, y) = 0$ for almost all $x \le y$. Consequently, it remains to verify the following lemma, which is proven below.

**Lemma D.2.** $\rho^c(x, y) = 0$ *for almost all* $x \le y$ *if and only if the null hypothesis of quasi independence is fulfilled.*

### D.1 Proof of Lemma D.1

First, observe that

$$\Psi_{c,n}^2 = \sup_{\omega \in B_1(\mathcal{H})} \left( \iint_{x \le y} \omega(x, y) \widehat{\rho}^c(dx, dy) \right)^2 = \|\phi_0^n - \phi_1^n\|_{\mathcal{H}}^2 = V_{0,0} - 2V_{0,1} + V_{1,1},$$

where

$$V_{0,0} = \|\phi_0^n\|_{\mathcal{H}}^2 = \frac{1}{n^2} \sum_{j,i=1}^{n} \mathfrak{K}((X_i, T_i), (X_j, T_j)) \Delta_i \Delta_j \widehat{\pi}^c(X_i, T_i) \widehat{\pi}^c(X_j, T_j),$$

$$V_{0,1} = \langle \phi_0^n, \phi_1^n \rangle_{\mathcal{H}} = \frac{1}{n^3} \sum_{i,j,k=1}^{n} \mathfrak{K}((X_i, T_i), (X_j, T_k)) \Delta_i \Delta_k \widehat{\pi}^c(X_i, T_i) \mathbb{1}_{\{X_k \le X_j < T_k \le T_j\}},$$

$$V_{1,1} = \|\phi_1^n\|_{\mathcal{H}}^2 = \frac{1}{n^4} \sum_{i,j,k,\ell=1}^{n} \mathfrak{K}((X_i, T_\ell), (X_j, T_k)) \Delta_k \Delta_\ell \mathbb{1}_{\{X_\ell \le X_i < T_\ell \le T_i\}} \mathbb{1}_{\{X_k \le X_j < T_k \le T_j\}}.$$

By Proposition A.1 we can replace $\widehat{\pi}_c$ by $\pi_c$ for all asymptotic considerations, a detailed explanation for $V_{0,0}$ is given below. Thus, $V_{0,0}$, $V_{0,1}$ and $V_{1,1}$ are asymptotically equivalent to $V$-statistics of order 2, 3 and 4, respectively. For the desired statement, it remains to show that (i) $V_{0,0} \to \|\phi_0\|_{\mathcal{H}}^2$ (ii) $V_{0,1} \to \langle \phi_0, \phi_1 \rangle_{\mathcal{H}}$ (iii) $V_1 \to \|\phi_1\|_{\mathcal{H}}^2$. All three convergences follow from the strong law of large numbers for $V$-statistics and Proposition A.1, as explained exemplary for (i):

Since the kernel $\mathfrak{K}$ and $\widehat{\pi}^c$ are bounded by some $c_1 > 0$ and 1, respectively, we can deduce from Proposition A.1 and the triangular inequality that almost surely

$$\left| \frac{1}{n^2} \sum_{j,i=1}^{n} \mathfrak{K}((X_i, T_i), (X_j, T_j)) \Delta_i \Delta_j \left( \widehat{\pi}^c(X_i, T_i) \widehat{\pi}^c(X_j, T_j) - \pi^c(X_i, T_i) \pi^c(X_j, T_j) \right) \right|$$

$$\le c_1 \frac{1}{n^2} \sum_{j,i=1}^{n} \Delta_i \Delta_j \left( |\widehat{\pi}^c(X_i, T_i) - \pi^c(X_i, T_i)| + |\widehat{\pi}^c(X_j, T_j) - \pi^c(X_j, T_j)| \right)$$

$$\le \frac{2c_1}{n^2} \sum_{j,i=1}^{n} \Delta_i \Delta_j \left| \widehat{\pi}^c(X_i, T_i) - \pi^c(X_i, T_i) \right|$$

$$\le \frac{2c_1}{n} \sum_{i=1}^{n} \Delta_i \left| \widehat{\pi}^c(X_i, T_i) - \pi^c(X_i, T_i) \right| \to 0.$$

Thus, we can replace for further asymptotic investigations $\widehat{\pi}^c$ by $\pi^c$. Finally, by the strong law of large numbers

$$V_{0,0} \to \mathbb{E}\Big( \mathfrak{K}((X_1, T_1), (X_2, T_2)) \Delta_1 \Delta_2 \pi^c(X_1, T_1) \pi^c(X_2, T_2) \Big)$$

$$= \iint_{x_1 < t_2} \iint_{x_2 < t_2} \mathfrak{K}((x_1, t_1), (x_2, t_2)) d\nu_0(x_1, t_1) d\nu_0(x_2, t_2).$$

## D.2 Proof of Lemma D.2

The first implication was already shown in the proof of Proposition 3.2. Now, assume that $\rho^c = 0$. Then

$$\pi^c(x,y)\frac{\partial^2 \pi_1^c(x,y)}{\partial x \partial y} = \frac{\partial \pi^c(x,y)}{\partial x}\frac{\partial \pi_1^c(x,y)}{\partial y}. \tag{18}$$

Define $M(x,y) = \frac{\partial \pi_1^c(x,y)}{\partial y} = \int_0^x S_{C|X=x'}(y)f_{XY}(x',y)dx'$, then Equation (18) can be rewritten as

$$\pi^c(x,y)\frac{\partial M(x,y)}{\partial x} = \frac{\partial \pi^c(x,y)}{\partial x}M(x,y). \tag{19}$$

Set $Q(x,y) = \mathbb{1}_{\{M(x,y)\neq 0\}}\pi^c(x,y)/M(x,y)$. From (18) we can conclude that $M(x,y) = 0$ implies $\pi^c(x,y) = 0$ or

$$0 = \frac{\partial^2 \pi_1^c(x,y)}{\partial x \partial y} = -S_{C|X=x}(y)f_{XY}(x,y).$$

But, the right-hand side of the equation is positive for all observable $(x,y)$, i.e. such that $S_{C|X=x}(y), f(x,y), f(x) > 0$. Note that only these pairs are relevant and, thus, we restrict to them subsequently. Thus, $\pi^c(x,y) = Q(x,y)M(x,y)$ and differentiation with respect to $x$ leads to

$$\begin{aligned}
\frac{\partial \pi^c(x,y)}{\partial x} &= \frac{\partial Q(x,y)}{\partial x}M(x,y) + Q(x,y)\frac{\partial M(x,y)}{\partial x}\\
&= \frac{\partial Q(x,y)}{\partial x}M(x,y) + \frac{\pi^c(x,y)}{M(x,y)}\frac{\partial M(x,y)}{\partial x}\\
&= \frac{\partial Q(x,y)}{\partial x}M(x,y) + \frac{\partial \pi^c(x,y)}{\partial x}.
\end{aligned}$$

Thus, $\partial Q(x,y)/\partial x = 0$ for all (observable) $x \leq y$. In particular, $Q$ does not depend on $x$, and we can write $Q(y)$ instead of $Q(x,y)$. Consequently, we can deduce from the definitions of $Q$, $M$ and $\pi^c$ that

$$-Q(y)\int_0^x S_{C|X=x'}(y)f_{XY}(x',y)dx' = \int_0^x S_{Y|X=x'}(y)S_{C|X=x'}(y)f_X(x')dx'.$$

In particular, we can deduce that for all observable $x \leq y$

$$-Q(y)S_{C|X=x}(y)f_{XY}(x,y) = S_{Y|X=x}(y)S_{C|X=x}(y)f_X(x).$$

From this we obtain

$$\begin{aligned}
f_{XY}(t,y) &= -Q(y)^{-1}S_{Y|X=t}(y)f_X(t)\\
\Leftrightarrow f_X(t)f_{Y|X=t}(y) &= -Q(y)^{-1}S_{Y|X=t}(y)f_X(t)\\
\Leftrightarrow \lambda_{Y|X=x}(y) &= -Q(y)^{-1},
\end{aligned}$$

where $\lambda_{Y|X=x}$ denotes the hazard rate function, which does not depend on $x$. Note that $S_{Y|X=x}(x) = 1$ and

$$S_{Y|X=x}(y) = \frac{S_{Y|X=x}(y)}{S_{Y|X=x}(x)} = \exp\Big(\int_x^y Q(s)^{-1}ds\Big).$$

Moreover, for $t < x < y$

$$\frac{S_{Y|X=x}(y)}{S_{Y|X=t}(y)} = \exp\Big(\int_x^y Q(s)^{-1}ds - \int_t^y Q(s)^{-1}ds\Big) = \frac{g(t)}{g(x)},$$

where $g(x) = \exp(\int_{l_X}^x Q(s)^{-1}ds)$ and $l_X = \inf\{s \geq 0 : f_X(s) > 0\}$ is the lower bound of the support of $X$ (given $X \leq Y$). Differentiation with respect to $y$ leads to

$$f_{Y|X=x}(y) = f_{Y|X=t}(y)\frac{g(t)}{g(x)}$$

and, thus,

$$f_{XY}(x,y) = \frac{f_{XY}(t,y)g(t)}{f_X(t)}\frac{f_X(x)}{g(x)}. \tag{20}$$

Now, let $(t_n)_{n\in\mathbb{N}}$ be a strictly decreasing sequence with $f(t_n) > 0$ and $t_n \to l_X$ as $n \to \infty$. Set $t_0 = \infty$. Then we can deduce from Equation (20) that

$$f_{XY}(x,y) = \widetilde{f}_Y(y)\widetilde{f}_X(x),$$

where

$$\widetilde{f}_Y(y) = \sum_{n=1}^{\infty} \frac{f_{XY}(t_n,y)g(t_n)}{f_X(t_n)}\mathbb{1}_{\{y\in(t_n,t_{n-1})\}}, \quad \widetilde{f}_X(x) = \frac{f_X(x)}{g(x)}.$$

# E Efficient implementation of wild bootstrap

Similarly to the work of [5], we can implement our Wild-Bootstrap efficiently by considering the identity $\text{trace}(\boldsymbol{AB}) = \sum_{ij}(\boldsymbol{A}\odot\boldsymbol{B})_{ij}$, where $\boldsymbol{A}$ and $\boldsymbol{B}$ denote $n\times n$ matrices, and $\odot$ denotes the element-wise product. By using this identity our test-statistic can be written as

$$\begin{aligned}
\Psi^2_{c,n} &= \frac{1}{n^2}\text{trace}(\boldsymbol{K}\widehat{\boldsymbol{\pi}}^c\widetilde{\boldsymbol{L}}\widehat{\boldsymbol{\pi}}^c - 2\boldsymbol{K}\widehat{\boldsymbol{\pi}}^c\widetilde{\boldsymbol{L}}\boldsymbol{B}^\mathsf{T} + \boldsymbol{K}\boldsymbol{B}\boldsymbol{L}\boldsymbol{B}^\mathsf{T}) \\
&= \sum_{ij}\left(\boldsymbol{K}\odot(\widehat{\boldsymbol{\pi}}^c\widetilde{\boldsymbol{L}}\widehat{\boldsymbol{\pi}}^c - 2\widehat{\boldsymbol{\pi}}^c\widetilde{\boldsymbol{L}}\boldsymbol{B}^\mathsf{T} + \boldsymbol{B}\boldsymbol{L}\boldsymbol{B}^\mathsf{T})\right)_{ij} \\
&= \sum_{ij}\boldsymbol{M}_{ij},
\end{aligned}$$

where $\boldsymbol{M} = \boldsymbol{K}\odot(\widehat{\boldsymbol{\pi}}^c\widetilde{\boldsymbol{L}}\widehat{\boldsymbol{\pi}}^c - 2\widehat{\boldsymbol{\pi}}^c\widetilde{\boldsymbol{L}}\boldsymbol{B}^\mathsf{T} + \boldsymbol{B}\boldsymbol{L}\boldsymbol{B}^\mathsf{T})$ is a $V$-statistic matrix. Then, the wild-bootstrap version of the preceding $V$-statistic is $(\Psi^{\text{WB}}_{c,n})^2 = W^\mathsf{T}\boldsymbol{M}W$ where $W = (W_1,\ldots,W_n) \in \mathbb{R}^n$ are the wild bootstrap weights. In this way, we only need to compute $O(n^2)$ sum once, for each wild bootstrap, instead of computing several (actually 6 times) $O(n^3)$ matrix multiplications and two $O(n^2)$ matrix multiplications for $K^W$.

# F Review of related quasi independence tests

In this section, we review the quasi-independence tests implemented in Section 5 of the main text.

**WLR** refers to the weighted log-rank test discussed in [7], which is defined as

$$L_W = \int_{x\leq y} W(x,y)\left\{N_{11}(dx,dy) - \frac{N_{1\bullet}(dx,y)N_{\bullet 1}(x,dy)}{R(x,y)}\right\},$$

where

$$N_{11}(dx,dy) = \sum_j \mathbb{1}(X_j = x, T_j = y, \Delta_j = 1), \quad N_{\bullet 1}(x,dy) = \sum_j \mathbb{1}(X_j \leq x, T_j = y, \Delta_j = 1),$$

$$N_{1\bullet}(dx,y) = \sum_j \mathbb{1}(X_j = x, T_j \geq y), \quad R(x,y) = \sum_j \mathbb{1}(X_j \leq x, T_j \geq y),$$

and $W : \mathbb{R}^2_+ \to \mathbb{R}$ is the weight function given by $W(x,y) = R(x,y)$. We note that, $R(x,y) = n\widehat{\pi}^c(x,y)$ defined in our notation. It is straightforward to see $\Psi^2_{c,n} = \frac{1}{n^2}L^2_W$ in the case $\mathfrak{K} = 1$.

**WLR_SC** refers to the previous log-rank test with weight $W$ given by $W(x,y) = \int_0^x \widehat{S}_{C_R}((y-u)-)^{-1}\widehat{\pi}^c(du,y)$, where $\widehat{S}_{C_R}$ is the Kaplan-Meier estimator based on the data $((C_i-X_i, 1-\Delta_i))_{i=1}^n$. tis specific test was proposed to the general assumption $Y_i \perp C_i|X_i$.

**M&B** refers to the conditional Kendall's tau statistic in discussed in [25]. Let

$$\begin{aligned}
B_{ij} &= \{\max(X_i, X_j) \leq \min(T_i, T_j)\} \\
&\cap \{(\Delta_i = \Delta_j = 1) \cup (T_j > T_i, \Delta_i = 1, \Delta_j = 0) \cup (T_i > T_j, \Delta_i = 1, \Delta_j = 0)\}.
\end{aligned}$$

The conditional Kendall's tau statistic is given by

$$\widehat{\tau}_b = \sum_{i<j} \mathbb{1}_{\{B_{ij}\}} \text{sign}((X_i - X_j)(T_i - T_j)).$$

**MinP1** and **MinP2** refers to the minimal p-value selection tests which are permutation based methods proposed in [3]. These tests are based on the underlying principle that, under quasi-independence, the distributions of $Y|X \leq t$ and $Y|X > t$ should not differ, where $t$ denotes some cut-point. Given a collection of possible cut-points $t$, the authors perform several two-sample log-rank tests for comparing $\{(T_i, \Delta_i) : X_i \leq t\}$ and $\{(T_i, \Delta_i) : X_i > t\}$ (under right-censored data), and set as their test-statistic the minimum log-rank $p$-value obtained. To guarantee meaningful comparisons, the authors consider cut-points that yield at least $E$ events in each group.

The first test proposed is the following:

**MinP1**:

1. Set $m = 0$
2. Set $m = m + 1$ and split the data into two groups $\{i : X_i \leq X_m\}$ and $\{i : X_i > X_m\}$.
3. Check the groups are admissible by verifying $E \leq \sum_{i=1}^{n} \Delta_i \mathbb{1}_{\{X_i < X_m\}} \leq n - E$. If the latter holds, perform a two-sample log-rank test for comparing $\{(T_i, \Delta_i) : X_i \leq X_m\}$ and $\{(T_i, \Delta_i) : X_i > X_m\}$, and record the $p$-value obtained. If the condition is not satisfied, record a $p$-value equal to 1.
4. If $m < n$ return to Step 2
5. Set as test-statistic $minp_1$ the smallest p-value obtained.

Alternatively, the authors propose a second test, which splits the data according to whether or not, the entry times belong to the interval $(t - \epsilon, t + \epsilon)$, where $t$, again, denotes a cut-point and $\epsilon > 0$. Similarly to the previous case, we need to ensure that each group contains at least $R$ data points, this can be done by choosing a suitable $\epsilon > 0$.

**MinP2**:

1. Set $m = 0$
2. Set $m = m + 1$ and split the data into two groups $\{i : X_i \in (X_m - \epsilon_m, X_m + \epsilon_m)\}$ and $\{i : X_i \notin (X_m - \epsilon_m, X_m + \epsilon_m)\}$, where $\epsilon_m$ is the smallest $\epsilon > 0$ such that there are at least $E$ data-points in each group. Record the value $\epsilon_m$.
3. If $m < n$ return to Step 2.
4. Set $\epsilon = \max_m \epsilon_m$ and $m = 0$
5. Set $m = m + 1$. Verify $E \leq \sum_{i=1}^{n} \Delta_i \mathbb{1}_{\{T_m - \epsilon < T_i < T_m + \epsilon\}} \leq n - E$ which checks that the partition of the data is admissible (under right-censoring). If the latter holds, perform a two-sample log-rank test for comparing each group and record the $p$-value. If the partition is not admissible record a $p$-value equal to 1.
6. If $m < n$ return to Step 5.
7. Set as test-statistic $minp_2$ the smallest p-value obtained.

The rejection regions for these tests are computed by using a permutation approach.

# G  Additional discussions for empirical results

In this section, we provide additional information and discussions on our empirical findings.

## G.1  Computational runtime

As shown in Table 3, our proposed test, implemented as described in Appendix E, has a significantly lower runtime when compared with the permutation approaches which require much longer run-time. M&B implements the conditional Kendall's tau statistic which has a closed-form expression for the null distribution, therefore the runtime is much lower again.

| n | 100 | 200 | 300 | 400 | 500 | 600 | 700 | 800 | 900 |
|---|---|---|---|---|---|---|---|---|---|
| KQIC | 0.012 | 0.019 | 0.031 | 0.041 | 0.063 | 0.085 | 0.130 | 0.152 | 0.200 |
| MinP1 | 15.77 | 41.62 | 56.61 | 90.52 | 113.7 | 154.4 | 254.4 | 299.2 | 389.1 |
| MinP2 | 20.33 | 35.08 | 59.09 | 101.4 | 123.7 | 174.3 | 242.4 | 300.9 | 354.2 |
| M&B | 0.002 | 0.002 | 0.002 | 0.003 | 0.004 | 0.006 | 0.006 | 0.009 | 0.021 |

Table 3: The runtime, in seconds, for a single trial using 500 wild bootstrap samples for KQIC and 500 permutations for MinP1 and MinP2. M&B does not require to approximate the null distribution.

## G.2  Kernel choice

**Parameter selection**  In kernel-based hypothesis testing, test power (i.e., the probability of rejecting $H_0$ when it is false) can vary for different choices of kernel parameters, such as the bandwidth in Gaussian kernels [17]. Previous works [17, 19, 20, 21, 32] have proposed to choose the kernel parameters by maximizing a proxy for the test power. In the uncensored setting, the test power is (to a good approximation) increased by maximising the ratio of the test statistic to its standard deviation under the alternative. We conjecture that the same ratio represents a good criterion in the setting of left-truncation and right-censoring, for which we have strong empirical evidence. A formal proof remains a topic for future work.

In the censored case, the test power criterion takes the form $\frac{\Psi_c^2}{\sigma_{H_1}}$, where $\sigma_{H_1}$ is the standard deviation of $\Psi_c^2$ under the alternative hypothesis $H_1$. Thus, to maximise the test power, we choose the kernel parameter $\theta$ by

$$\theta^* = \arg\max_{\theta} \frac{\Psi_c^2}{\sigma_{H_1}}.$$

In practice, we use part of the data to compute $\Psi_{c,n}^2/(\widehat{\sigma}_{H_1} + \lambda)$, where $\widehat{\sigma}_{H_1}$ is an empirical estimate of $\sigma_{H_1}$ and a regularisation parameter $\lambda > 0$ is added for numerical stability. We then perform the test on the remaining data with the selected $\theta^*$. A $20/80$ train-test split is suggested in [21] for learning the parameter. We use the regulariser $\lambda = 0.01$.

We next give our empirical estimate for the variance $\widehat{\sigma}_{H_1}^2$. First, $\Psi_{c,n}^2$ can be written as $\Psi_{c,n}^2 = \frac{1}{n^2} \sum_{i=1}^{n} \sum_{j=1}^{n} J_n((T_i, \Delta_i, X_i), (T_j, \Delta_j, X_j))$, where $J_n$ is defined by

$$J_n((T_i, \Delta_i, X_i), (T_j, \Delta_j, X_j)) = \Delta_i \Delta_j L(T_i, T_j) g_n(X_i, X_j),$$

where

$$g_n(X_i, X_j) = K(X_i, X_j)\widehat{\boldsymbol{\pi}}^{\boldsymbol{c}}_{ii}\widehat{\boldsymbol{\pi}}^{\boldsymbol{c}}_{jj} - 2\sum_{l=1}^{n} K(X_i, X_l)\widehat{\boldsymbol{\pi}}^{\boldsymbol{c}}_{ii}\boldsymbol{B}_{l,j} + \sum_{l=1}^{n}\sum_{k=1}^{n} K(X_k, X_l)\boldsymbol{B}_{k,i}\boldsymbol{B}_{l,j},$$

and $\widehat{\boldsymbol{\pi}}^{\boldsymbol{c}}_{ii} = \widehat{\pi}^c(X_i, T_i)$ and $B_{k,i} = \mathbb{1}_{\{X_i \le X_k < T_i \le T_k\}}/n$. This "$V$-statistic" form suggests that the variance can be estimated by

$$\widehat{\sigma}_{H_1}^2 = \frac{1}{n}\sum_{i=1}^{n}\left(\frac{1}{n}\sum_{j=1}^{n} J_n(i,j)\right)^2 - \left(\frac{1}{n^2}\sum_{i=1}^{n}\sum_{j=1}^{n} J_n(i,j)\right)^2,$$

where $J_n(i,j) = J_n((T_i, \Delta_i, X_i), (T_j, \Delta_j, X_j))$.

Finally, some remarks on the performance of our kernel selection heuristic in experiments. For simple cases, our kernel selection procedure makes little difference, since a broad range of kernel bandwidths yields good results, and the "median heuristic" (selection of the bandwidth as the pairwise inter-sample distance) is adequate. On the other hand, our procedure results in large power improvements for more complex cases such as periodic dependency at high frequencies, where the median distance between samples does not correspond to the lengthscale at which dependence occurs. Similar phenomena have also been observed previously in [32].

**Inverse Multi-Quadratic (IMQ) kernel** We further study the performance of the IMQ kernel on our proposed test. The IMQ kernel has the form $k(x, y) = (c^2 + \|x - y\|^2)^b$, for constant $c > 0$ and $b \in (-1, 0)$. As proposed in [12], we choose $b = -\frac{1}{2}$. We select the parameter $c$ by maximizing a heuristic proxy for test power, as discussed above. The controlled Type-I error is shown in Table 4, where $X$ and $Y$ are independent samples from $\mathrm{Exp}(1)$. Truncation and right-censoring apply with censoring time independently generated from exponential distribution. We report the test power of KQIC with IMQ kernel in later sections.

| n | 50 | 100 | 150 | 200 | 250 | 300 | 350 | 400 | 450 | 500 |
|---|---|---|---|---|---|---|---|---|---|---|
| KQIC_IMQ | 0.08 | 0.05 | 0.03 | 0.05 | 0.04 | 0.05 | 0.05 | 0.07 | 0.07 | 0.05 |

Table 4: Type-I error for IMQ kernels, with $\alpha = 0.05$, censoring level $25\%$, 100 trials, and increasing sample size $n$.

## G.3 Periodic dependencies

As briefly mentioned in the main text, the parameter $\beta$ controls the frequency of sinusoidal dependence. At a given sample size, dependence becomes harder to detect as the frequency $\beta$ increases, both for our test and for competing methods. We illustrate the datasets visually in Figure 7. For a fixed sample size, the test power decreases as frequency increases, which is observed in our results in Figure 4. For high frequency cases, larger sample size is required to correctly reject the null as shown in Figure 5.

Figure 7: Samples from periodic dependency model w.r.t. frequency coefficient $\beta$.

(a) Sample size: n = 50

(b) Sample size: n = 100

(c) Sample size: n = 300

Type-I error is reported in Table 5, and is close to the desired level (subject to finite sample effects).

## G.4 Dependent censoring

In this section we show that our test achieves correct Type-I error under the null hypothesis even when considering dependent censoring times $C$. As stated in Assumption 3.1, we only require $Y \perp C | X$, which is a standard assumption, as also considered in [7].

| n | 100 | 300 | 500 | 700 | 900 | 1100 | 1300 | 1500 | 1700 | 1900 |
|---|---|---|---|---|---|---|---|---|---|---|
| KQIC_Gauss | 0.045 | 0.060 | 0.055 | 0.040 | 0.045 | 0.045 | 0.040 | 0.030 | 0.045 | 0.050 |
| KQIC_IMQ | 0.050 | 0.055 | 0.045 | 0.030 | 0.020 | 0.040 | 0.025 | 0.020 | 0.015 | 0.020 |
| WLR | 0.030 | 0.045 | 0.050 | 0.025 | 0.045 | 0.015 | 0.015 | 0.030 | 0.025 | 0.040 |
| WLR_SC | 0.035 | 0.060 | 0.030 | 0.025 | 0.060 | 0.070 | 0.045 | 0.055 | 0.050 | 0.060 |

Table 5: Type-I error with $\alpha = 0.05$, censoring level $25\%$, 200 trials, and increasing sample size $n$.

Figure 8: Samples generated from $H_0$ with periodic dependent censoring distributions.

We generate the data as follows: Sample $X_i \sim \mathrm{Exp}(1)$, then generate $Y_i \sim \mathrm{Exp}(1)$ (independent of $X_i$) and $C_i | X_i \sim \mathrm{Exp}(e^{\cos(2\pi\gamma X_i)})$. Generate the observed data point $(T_i, \Delta_i, X_i)$, where $T_i = \min\{Y_i, C_i\}$ and $\Delta_i = \mathbb{1}_{\{T_i = Y_i\}}$ and keep it as a valid sample only if $T_i \geq X_i$. Notice that in this case both left truncation and right-censoring are present in the data. Also, notice that the null hypothesis holds since the survival times $Y_i$ are quasi-independent of the entry times $X_i$. In Figure 8, we show the unobserved pairs $(X, Y)$ and the observed pairs $(X, T)$ where the censoring variable is generated using different censoring frequencies $\gamma$. From the plot, we see that the entry times $X$ and survival times $Y$ look quasi-independent, but, due to the periodic dependency of the censoring distribution, the observed data $(X, T)$ show a periodic trend, which looks similar to the observations in Figure 7. However, since this dependency is due to the censoring times $C$ instead of the survival times $Y$, our tests are able to recover $H_0$ and achieve correct test level, as shown in Table 6. The tests proposed in [7] are also valid under Assumption 3.1, thus we include the results for WLR and WLR_SC as well. From Table 6, we observe that KQIC with both Guassian and IMQ kernels, as well as WLR achieve the correct test level; however, WLR_SC has slightly higher type-I errors when sample size is small and achieves correct test-level when sample size becomes large (recall that WLR_SC uses a data dependent weight, thus convergence in this case might be slower).

### G.5 Test performance w.r.t. censoring level

We report the Type-I error for different censoring percentages, see Table 7. With reasonable censoring level (e.g. $< 90\%$), the Type-I errors are well controlled. WLR_SC has higher Type-I with small sample sizes, which is similarly observed in Table 6. However, the Type-I error is less controlled at extremely high censoring percentages, due to the lack for useful information obtained. In practise, we may need to be careful dealing with extremely high censoring when applying the quasi-independence tests.

Table 6: Type-I error for periodic dependent censoring distributions, with $\alpha = 0.05$ and 100 trials.

| n | 100 | 200 | 300 | 400 | 500 | 600 | 700 | 800 | 900 | 1000 |
|---|---|---|---|---|---|---|---|---|---|---|
| KQIC_Gauss | 0.07 | 0.06 | 0.03 | 0.03 | 0.06 | 0.05 | 0.04 | 0.04 | 0.03 | 0.07 |
| KQIC_IMQ | 0.07 | 0.06 | 0.04 | 0.01 | 0.03 | 0.04 | 0.05 | 0.05 | 0.06 | 0.07 |
| WLR | 0.07 | 0.05 | 0.03 | 0.01 | 0.03 | 0.04 | 0.05 | 0.04 | 0.03 | 0.07 |
| WLR_SC | 0.10 | 0.08 | 0.09 | 0.13 | 0.13 | 0.04 | 0.09 | 0.05 | 0.04 | 0.06 |

(a) Censoring frequency $\gamma = 0.5$. Censoring level 30%

| n | 100 | 200 | 300 | 400 | 500 | 600 | 700 | 800 | 900 | 1000 |
|---|---|---|---|---|---|---|---|---|---|---|
| KQIC_Gauss | 0.03 | 0.02 | 0.01 | 0.04 | 0.05 | 0.06 | 0.05 | 0.04 | 0.06 | 0.04 |
| KQIC_IMQ | 0.02 | 0.03 | 0.03 | 0.03 | 0.04 | 0.05 | 0.05 | 0.04 | 0.04 | 0.04 |
| WLR | 0.02 | 0.02 | 0.03 | 0.05 | 0.04 | 0.04 | 0.04 | 0.04 | 0.05 | 0.05 |
| WLR_SC | 0.06 | 0.12 | 0.17 | 0.15 | 0.10 | 0.11 | 0.06 | 0.04 | 0.05 | 0.05 |

(b) Censoring frequency $\gamma = 1.2$. Censoring level 35%

| n | 100 | 200 | 300 | 400 | 500 | 600 | 700 | 800 | 900 | 1000 |
|---|---|---|---|---|---|---|---|---|---|---|
| KQIC_Gauss | 0.06 | 0.04 | 0.06 | 0.02 | 0.02 | 0.04 | 0.03 | 0.03 | 0.05 | 0.04 |
| KQIC_IMQ | 0.05 | 0.04 | 0.05 | 0.01 | 0.02 | 0.04 | 0.04 | 0.03 | 0.03 | 0.04 |
| WLR | 0.04 | 0.02 | 0.04 | 0.02 | 0.02 | 0.04 | 0.04 | 0.03 | 0.05 | 0.03 |
| WLR_SC | 0.09 | 0.10 | 0.13 | 0.15 | 0.10 | 0.08 | 0.05 | 0.03 | 0.03 | 0.04 |

(c) Censoring frequency $\gamma = 3.0$. Censoring level 40%

| % censored | 20 | 35 | 50 | 70 | 85 | 92 | 95 |
|---|---|---|---|---|---|---|---|
| $n = 200$ | | | | | | | |
| KQIC_Gauss | 0.040 | 0.025 | 0.015 | 0.045 | 0.035 | 0.085 | 0.115 |
| KQIC_IMQ | 0.040 | 0.060 | 0.050 | 0.055 | 0.070 | 0.100 | 0.185 |
| WLR | 0.055 | 0.035 | 0.040 | 0.050 | 0.030 | 0.075 | 0.120 |
| WLR_SC | 0.045 | 0.105 | 0.075 | 0.120 | 0.060 | 0.035 | 0.075 |
| $n = 300$ | | | | | | | |
| KQIC_Gauss | 0.055 | 0.040 | 0.055 | 0.045 | 0.060 | 0.090 | 0.065 |
| KQIC_IMQ | 0.045 | 0.050 | 0.070 | 0.050 | 0.050 | 0.105 | 0.115 |
| WLR | 0.030 | 0.055 | 0.055 | 0.040 | 0.050 | 0.075 | 0.065 |
| WLR_SC | 0.080 | 0.120 | 0.140 | 0.095 | 0.125 | 0.095 | 0.025 |
| $n = 500$ | | | | | | | |
| KQIC_Gauss | 0.040 | 0.050 | 0.035 | 0.030 | 0.030 | 0.050 | 0.090 |
| KQIC_IMQ | 0.065 | 0.030 | 0.050 | 0.040 | 0.080 | 0.100 | 0.050 |
| WLR | 0.035 | 0.035 | 0.050 | 0.035 | 0.040 | 0.060 | 0.075 |
| WLR_SC | 0.060 | 0.035 | 0.055 | 0.075 | 0.065 | 0.035 | 0.015 |
| $n = 800$ | | | | | | | |
| KQIC_Gauss | 0.045 | 0.030 | 0.030 | 0.065 | 0.030 | 0.065 | 0.080 |
| KQIC_IMQ | 0.065 | 0.050 | 0.050 | 0.060 | 0.060 | 0.090 | 0.140 |
| WLR | 0.015 | 0.010 | 0.025 | 0.055 | 0.065 | 0.085 | 0.100 |
| WLR_SC | 0.095 | 0.040 | 0.065 | 0.080 | 0.075 | 0.045 | 0.025 |

Table 7: Type-I error for different censoring level, with $\alpha = 0.05$ and 200 trials,