[Reviews · NeurIPS 2020]

Review 1

Summary and Contributions: This paper proposes a test of quasi-independence (i.e. are X and Y independent, apart from the fact that, by design, X < Y), that applies in the right-censored setting. The new test consists in considering a generalization of a log rank test, where instead of having \omega fixed, the supremum over a rkhs ball is taken.

Strengths: - the paper is clear and well written - the mathematical aspects are correctly dealt with - the applications targeted are clearly identified

Weaknesses: - However, I found the overall contribution slightly insufficient for acceptance, as it focuses on a somewhat specific application (quasi independence under right-censoring) - In particular, the differences between Sections 2 and 3 are not highlighted enough in my point of view, I almost had the feeling to read twice the same work - Can authors motivate the choice of the factorising kernel, it seems a bit restrictive to decouple the effect on x and y? - Can one imagine a kernel directly dealing with the censoring instead of considering the ^c functions? - Censoring data can be seen as biased observations, coming from the original distribution biased through the multiplication by the indicator function 1{ Y < C}. Can more complex biasing functions be considered in the proposed analysis, and why? Possibly applying on x? - The experiments on real data are not so stellar

Correctness: Yes, I did not carefully check all mathematical proofs

Clarity: Yes

Relation to Prior Work: Yes, but the novelty seems a bit insufficient to me

Reproducibility: Yes

Additional Feedback: Response to rebuttal: I thank the authors for their feedback, that partially answered my concerns. I will not fight acceptance, and have updated my score to 5.


Review 2

Summary and Contributions: The quasi-independence (QI) model has been extensively studied, first in the contingency table literature and an extension to the continuous case, for truncated data, was given by Tsai (1990). This was further developed by Emura and Wang (2010). The present contribution builds forth on this work and develops a kernel test. An asymptotic analysis is given and a wild bootstrap is proposed.

Strengths: The proposed kernel test seems to be a statistically natural and mathematically elegant way to test for QI. The approach is quite general and it seems to me supersedes previous work. The methodology is complete, giving both extensive theory and a testing algorithm.

Weaknesses: None

Correctness: Yes

Clarity: Yes

Relation to Prior Work: Yes

Reproducibility: Yes

Additional Feedback: It took me a while to understand exactly what was meant in the paper by QI. In the abstract, it says "...it is still of interest to determine whether there exists significant dependence beyond their ordering in time", a sentence I do not quite follow. On first reading, I thought why not look at the dependence between X and Y-X. But the definition is given in (1), which if I understand correctly, means that if we have two (possibly hypothetical) independent variables X and Y, and only observe (X,Y) such that X<Y, then (1) follows. I felt some motivation for (1) can be given, a short derivation might be instructive. (I did get confused a bit initially that there was no conditioning on X<Y, but I understood after looking at Emura and Wang and the remark in line 98.) Response to author feedback: I am satisfied with the response.


Review 3

Summary and Contributions: This paper considers observed data with (left-) truncation and (right-) censoring, and establishes kernel-based nonparametric tests of quasi-independence, i.e., the non-existence of dependence beyond the temporal ordering in which the variables are observed. The proposed approach is shown to recover existing parametric approaches as special cases, and experiments demonstrate the the test attains higher test power than existing approaches without sacrificing computational efficiency.

Strengths: The paper addresses an interesting and practical problem, and proposes a nonparametric and theoretically justified approach. While there have been recent works on kernel-based two-sample and goodness-of-fit tests for censored data, the truncation setting studied in the current work poses additional challenges, and I believe the proposed quasi-independence tests could be of use to practitioners.

Weaknesses: The proposed methodology is specifically designed for data with truncation and censoring, as existing approaches such as the HSIC have already addressed the more general scenarios. In practice, one major limitation of the proposed approach is that in such data with truncation and censoring, it may be likely that some form of even quasi-dependence would have crept into the data as e.g., caused by the underlying study design etc, which would (ideally) cause the test to reject the null-hypothesis. Thus, a more useful question might be to quantify the degree of quasi-independence and the extent to which it might affect downstream analyses, rather than a binary accept/reject decision. But of course, the paper presents an appealing first step in this direction.

Correctness: Yes, to the best of my knowledge. I have skimmed through the supplementary material, but have not verified the proofs rigorously.

Clarity: Yes, the paper is clearly written.

Relation to Prior Work: Yes, the paper provides extensive discussion to existing parametric tests. I also find the review of the existing tests in the supplementary material helpful.

Reproducibility: Yes

Additional Feedback: EDIT: I thank the authors for their response, which have addressed my questions. I leave my evaluation unchanged. ============================================= I have a few specific comments/questions: - For equation (1), what are interesting/representative cases where other forms of F_X and S_Y are used beyond the common CDF and survival functions? - In line 103, upon first reading, it’s not immediately clear how the expression relates to log-rank test-statistics which are more commonly known in discrete cases. It might be helpful to briefly elaborate on the connection. - In selecting/defining the kernel function, does the special care need to be taken to account for the underlying constrained spaced implied by the temporal ordering? - While some runtime results are included in the supplementary material, it would be helpful to have a brief discussion/comparison of the theoretical computational complexities (big-O) of computing the test statistic for both the proposed and existing approaches. - In experiments, how do WLR and WLR_SC fare in terms of computational cost?


Review 4

Summary and Contributions: The authors propose a nonparametric statistical test of quasi-independence which can be applied in the right-censored setting and has several real-life applications. They also provide an asymptotic analysis of the test statistic and demonstrate in experiments that it obtains better power than existing approaches while being more computationally efficient.

Strengths: EDIT: I thank the authors for their response, which have addressed my questions. I will keep the same score. -------------------------------------------- There are several strengths to this paper. 1. The work is well-grounded in the theoretical perspective. 2. The empirical evaluation is strong where they have applied it challenging synthetic problems as well as real-world examples. 3. The method extends from the previous work [9] where they show a nonparametric generalization. 4. The method is highly relevant to the NeurIPS community due to its broad applications.

Weaknesses: 1. My only concern is that this is a highly theoretical paper and most of the details are kept in the appendix. Due to the nature of the conference it might be hard for readers to go through such details. I would urge the authors to submit the full version to a journal. 2. The code does not open-sourced. It might be good if the authors could release the code and the datasets for ease of reproducibility.

Correctness: The claims and empirical methodology seems to be correct, though I have not gone through the complete details in the Appendix

Clarity: The paper is very well written and is very easy to follow.

Relation to Prior Work: The paper contains a very good coverage of related literature and explicitly mention how they extend on it.

Reproducibility: Yes

Additional Feedback:

[Author Response · NeurIPS 2020]

We thank all the reviewers for their helpful comments and feedback.

*To Reviewer 1* **Relevance of application domains; distinctions between Sec.2 and 3.** Quasi-(in)dependent data,
where one event occurs after another, are very common in many fields, e.g., biomedical studies, social sciences,
marketing analyses, etc. A test of Quasi-Independence (QI) thus has a broad range of practical applications.

QI and Right-Censoring (RC) are very different data properties. Firstly, QI is a deterministic hard constraint ($X \leq Y$),
while RC is a stochastic property of the data (incomplete observations). In this paper we solve the problem of testing
QI, giving a general solution in Sec.2. Additionally, since many real-life scenarios display RC (especially biomedical),
we extend our approach (given in Sec.2) to handle RC data (Sec.3). We will better emphasise the distinction in the final
version.

**Why a factorizing kernel?** There are two reasons: (1) This choice is sufficient to solve the problem: our resulting $\mathfrak{K}$
is $c_0$-universal, and Theorem 4.2 shows power goes to 1 asymptotically, meaning our test correctly rejects the null
hypothesis for any dependency (with enough data). This is illustrated in our experiments on complex scenarios. (2) The
resulting test statistic has a simple expression, Equation (7), which leads to a computationally efficient test.

**Using a kernel that directly incorporates censoring?** This is an interesting question. Note that a deterministic kernel
(such as ours) has no way of encoding random censoring. Thus, in order to allow the kernel to correct the bias due to
RC, we would need to consider a random kernel. Random kernels are much harder to analyse, thus we leave this as a
potential future research direction.

**Bias from censoring and other form of censoring?** First a point of clarification: our test-statistic uses *all* the data
$(X, T, \Delta)$, and not only observations biased by the multiplication by the indicator function, i.e, $(X\mathbf{1}_{\{Y<C\}}, T\mathbf{1}_{\{Y<C\}})$.
That said, we would be very interested in extending our work to other forms of censoring, such as interval censoring.

**Experiments on real data.** We believe that the real data experiments show an exciting result: on the abortion dataset,
with a very high level of censoring (90%), our test is the only one to correctly reject the null, and detect the signal in the
presence of such heavy censoring. This strong performance under heavy censoring is further confirmed in our appendix
G, fig. 8, showing on synthetic data that we can correctly detect quasi-dependence even with high censoring levels.

*To Reviewer 2* **Dependency beyond ordering.** Thanks for your suggestions for clarifying the presentation of the QI
setting. We will include a more detailed explanation, and some examples illustrating the problem.

*To Reviewer 3* **Quantify degree of quasi-dependency instead of testing?** We agree that a measure of the degree of
quasi-dependence would be very valuable, and an interesting research direction. There remain settings where a binary
decision can be important, however: in particular, QI is associated with simpler models, which can be employed if the
test accepts the null, but not if the null is rejected. This is the case for [28], where (unlike prior works) the authors don't
ignore quasi-dependence, and obtain completely different results.

**Alternative forms of $F_X$ and $S_Y$ can be used?** While Equation (1) formally defines quasi-independence, deriving the
specific form of $\widetilde{F}_X$ and $\widetilde{S}_Y$ can be complex, and will depend heavily on the specific problem setting; see the proof of
Lemma D.2 in the supplement.

**Link to the log-rank test.** Log-rank tests are best known in the two-sample test setting, however they can be easily
extended to other settings (continuous time, covariates, etc), by exploiting their relationship with score tests. This
general idea was used by Emura and Wang [9] in obtaining the term inside the parenthesis of Equation (4), known as
the weighted log-rank statistic (with weight function $\omega$).

**Kernel choice with temporal ordering.** The kernel does not need to be designed to incorporate the constraints implied
by the temporal ordering, or by RC. All the information about the data is fed through the functions $\rho$ and $\rho^c$, which
impose the relevant constraints arising from order and RC. The kernel defines the space of functions $\omega$ used to test the
null hypothesis. We use a $c_0$-universal (Theorem 4.2) kernel, for which the space is rich enough that the power goes to
1 asymptotically for any alternative.

**Computational complexity.** To compute the kernel based statistics of sample size n, it takes $O(n^3)$ with efficient
computation of $\hat{\pi}$ and $A$, $B$ matrices. A testing procedure with m wildboostraps takes $O(mn^2)$ to compute. Thus,
overall cost is $O(n^3 + mn^2)$.

**Computational cost comparison of WLR and WLR_SC.** The weighted log-rank test can be seen as a special form
of the proposed test, with a constant-valued kernel. Computing WLR and WLR_SC takes $O(n^3)$ (same as KQIC).
Instead of using wild bootstrap for the test threshold, however, Emura and Wang [9] suggests obtaining the rejection
threshold by approximating the relevant integrals empirically, which also costs $O(n^3)$. The overall cost is then $O(n^3)$.

*To Reviewer 4* **Open-source code.** We will provide a link to the source code in the final version of the paper.

[Meta-Review · NeurIPS 2020]

Three excellent reviewers felt that this was a very good contribution, and while the fourth was slightly negative, I read the paper myself and concur that testing for quasi-independence (independence in the presence of a known ordering effect) is an interesting problem, and handling right-censored data is critical in clinical trials. The kernel methodology is solid, and this will have several applications. One drawback is the lack of an interesting (nontrivial) power analysis, but that is common to most papers in the kernel testing literature. Overall I congratulate the authors on a job well done.